



# A Comprehensive Design Methodology of Shared Mooring Line Configurations for Assessing Mooring Costs and Performances of Floating Offshore Wind Turbines

Qi Pan[1], Dexing Liu[1], Feng Guo[2], and Po Wen Cheng[1]

[1]Stuttgart Wind Energy, University of Stuttgart, Allmandring 5b, 70569 Stuttgart, Germany
[2]State Key Laboratory of Ocean Engineering,Shanghai Jiao Tong University, China

**Correspondence:** Qi Pan (pan@ifb.uni-stuttgart.de)

**Abstract.**

Shared moorings possess the potential to create more cost-effective designs for large-scale deployments of floating wind farms than conventional mooring configurations. Existing stiffness linearization methods are valuable for conceptual designs, but are limited by the assumption of small floater offsets, which can lead to impractical mooring designs. Additionally, current

studies lack consensus on determining offset limits. To address these limitations and improve the reliability of shared mooring designs, this paper presents a comprehensive design methodology. This approach synthesizes realistic line characteristics, considers nonlinear mooring stiffness, incorporates multiple design constraints, and integrates wind-farm layouts to achieve diverse design objectives. By employing the design method, both conventional and shared mooring designs are generated, and the cost-saving trend of shared moorings is revealed. The mooring cost distribution for shared mooring designs skews towards

lower costs compared to conventional designs, showing 11% and 14% savings in the maximum material costs, respectively, for the two-turbine and three-turbine arrays in water depths of 200 m. This updates the findings of existing linear methods that shared moorings can be cost-effective in water depths exceeding 400 m. In addition, dynamic simulations demonstrate that the distance-driven shared mooring line configuration improves the mean power production by 2.3% for the downstream turbine in the three-turbine array, despite a notable increase in the horizontal offset of the downstream model and a shortened mooring

fatigue life of the upstream model. This paper provides insights into shared mooring designs and highlights the advantages of shared mooring designs in terms of material cost saving and power improvement, as well as the challenges related to platform motion and mooring line fatigue. These findings contribute to the development of cost-effective mooring solutions for floating wind projects.

## 1   Introduction

The year of 2023 has witnessed a total 8.8 GW of new installations, marking a 16% year-on-year growth in the global offshore wind power capacity (GWEC, 2023). However, challenges on rising capital costs are currently pushing back the advancement of floating offshore wind technology. As a result, it is predicted that the floating offshore wind market will not achieve full commercialization until 2030 (GWEC, 2023). Nevertheless, in the long term, the floating wind market demonstrates huge





opportunities, with ambitious goals to scale up from handful prototype projects to the GW-scale generating capacity. Technical
solutions that mitigate risk and reduce costs can help to achieve this goal.

Compared to bottom-fixed offshore turbines, floating offshore wind turbines (FOWTs) entail additional expenses for mooring
systems. The sensitivity analysis of the levelized cost of energy (LCOE) for floating wind farms highlights that LCOE is most
sensitive to manufacturing costs, including those of the wind turbine, the substructure and the mooring system (Lerch et al.,
2018). Therefore, reducing the material cost of the mooring system is crucial for achieving a cost-effective solution for floating
wind farms. The concept of shared mooring is proposed as an innovative technology (Beiter et al., 2016) and holds the potential
to decrease mooring material costs by reducing the length of mooring lines or the number of anchors.

Two concepts of shared mooring designs have been developed for FOWTs, utilizing either a shared anchor or a shared line
for multiple FOWTs. The first concept, shared anchor design, employs a single anchor to secure multiple mooring lines. This
design has been implemented in Hywind Tampen project (Equinor, 2023), where 11 FOWTs were fastened by 19 anchors
instead of the conventional 33 anchors (BVG Associates Ltd, 2023). The second concept, shared mooring line design, is
studied in this paper. Compared to the conventional design that consists of three lines and three anchors, a shared mooring line
configuration can potentially reduce both the number of anchors and the length of mooring lines, by connecting two floaters
without anchoring the line to the seabed. These reduction contributes to minimizing the material costs of floating wind farms.

Exploring the cost-saving potential of shared mooring line designs is critical to promoting their applications in floating wind
farms. Connolly and Hall (2019) estimated the material and installation costs of mooring lines and drag embedment anchors
for the NREL 5-MW FOWT model across different water depths from 200 m to 800 m. Their study demonstrated that shared
moorings could yield cost-saving benefits for FOWTs in water depths exceeding 400 m. However, at the early study phase
of shared mooring line configurations, it is difficult to estimate the installation costs of mooring systems for floating wind
farms across different water depths. Additionally,  limited research has quantified the cost benefits of shared mooring line
configurations compared to conventional mooring designs for floating wind turbines, particularly concerning mooring material
costs, which play a more significant role in LCOE than installation costs (Lerch et al., 2018).

The shared mooring line configurations introduce additional complexity to floating wind turbine arrays compared to con-
ventional mooring designs. This complexity arises because multiple FOWTs are coupled through shared lines, and the shared
line can alter the mooring stiffness of the sea-keeping system, thereby essentially affecting the global response of connected
FOWTs. Various studies underscore diverse influences of shared mooring designs on floater performances. Liang et al. (2020)
conducted a static analysis of two spar-type FOWTs with the NREL 5-MW wind turbine and a shared mooring line design.
They observed that static mooring restoring forces in surge and sway directions were relatively insensitive and strongly sensi-
tive to the shared mooring configuration, respectively. Similarly, Gözcü et al. (2022) analyzed the IEA 15-MW FOWT model
with a shared mooring line configuration, and noted a significant change in floater surge excursion with varying shared line
lengths. In addition, dynamic simulations were performed to analyze the responses of FOWTs with shared line configurations.
Munir et al. (2021) studied two 5-MW semi-submersible FOWTs with shared mooring line configurations, which resembled
that used by Liang et al. (2020) and Liang et al. (2023), where the shared line is suspended above the seabed with some clear-
ance, and found that the standard deviation of the floater sway motion increased by up to 50% with the use of a shared mooring





configuration. They also observed a minimal effect of shared line length on floater motions, contrary to observations from
static tests by Gözcü et al. (2022), where the shared line was partially laid down on the seabed. These varied observations from
the literature on shared line configurations highlight the importance of shared mooring designs. However, currently, there is a
lack of a comprehensive design methodology for shared mooring line configurations, as existing studies only consider limited
design parameters or constraints.

Existing design approaches rely on the assumption of small floater movements and implement linear spring models for
mooring lines. Connolly and Hall (2019) modeled the mooring line as a linear spring and chose the floater's offset under the
constant wind force as the only design constraint. In each shared mooring layout, the only design parameter, namely the mass
density of the shared line, was scaled linearly to fit the requirement of floater's offset. Similarly, Wilson et al. (2021) proposed
a linearized design method for shared mooring layouts, where each line was modeled as a linear horizontal spring. This design
approach was also adopted in subsequent works by Hall et al. (2022) and Lozon and Hall (2023). The horizontal movement of
the floater, serving as the only design constraint, was calculated by dividing the constant wind force by the horizontal stiffness
matrix. These linear methods can be useful in conceptual designs of shared mooring layouts. However, the linear scaling of
mooring line mass density and the absence of realistic chain properties, such as, nonlinear mooring stiffness arising from line
geometry and elasticity, may result in impractical designs of shared mooring lines. For example, in shared mooring layouts at
the water depth of 200 m, the minimum mass density of shared lines was reported as 26 kg/m (Connolly and Hall, 2019). And
in the shared mooring layouts for 3 turbines, where each turbine was supported by two shared lines and one non-shared line,
the maximum mass density of mooring lines reached 872 kg/m (Wilson et al., 2021). These two mass densities are beyond the
typical range of mass densities observed in mooring lines that used in existing floating projects.

Moreover, the reliability of mooring designs derived through these linear methods fully depends on an accurate assessment
of the floater's offset. This offset limit is critical for dynamic cable design, as excessive floater displacements could increase the
curvature of the cable. However, current research lacks consensus to determine its value for shared mooring designs. Connolly
and Hall (2019) used a fixed offset of 20 m for mooring designs across four different water depths ranging from 200 m to 800
m. Wilson et al. (2021) chose 10% of water depth, equal to 60 m, as the floater's offset to calculate the linearized mooring
stiffness, and 12% of water depth to adjust the line length. But field observations have proven that dynamic cables remained
safe when the 100-kW FOWT moved up to 21 m during the typhoon seasons, at locations with water depths ranging from 80
m to 100 m (Taninoki et al., 2017). Meanwhile, the simulations of the 2-MW FOWT at the same site showed that the dynamic
cable can operate safely even when the motion of the floater reached 40 m (Taninoki et al., 2017). These findings raise doubts
about the validity of assuming small floater movements and the reliability of the linearized shared mooring design methods
that consider the floater movement as the only design constraint. Therefore, shared mooring line configurations require a more
integrated and robust design framework that explores a variety of line and chain parameters and adapts to different practical
constraints, rather than merely adhering to the uncertain offset constraint of dynamic cable designs.

To address the limitations of existing linearized design method and to improve the reliability of shared mooring designs, this
paper proposes a comprehensive design methodology for shared mooring line configurations. This methodology synthesizes
various cable characteristics, considers nonlinear mooring stiffness, incorporates multiple design constraints, and integrates





the wind farm layout to achieve different design goals. The cost-driven design that minimizes mooring material costs, and the distance-driven design that shortens the distance between upstream and downstream turbines are analyzed in this paper. In addition, a material cost model is adopted to quantify cost-saving benefits of shared moorings over conventional mooring designs. Furthermore, the effects of shared mooring line configurations on the power production, floater motion and mooring tension fatigue for the FOWT models are assessed in this paper. Dynamic aero-hydro-servo-elastic simulations are performed for the floating turbine arrays with both cost-driven and distance-driven mooring configurations. The simulation results are compared to evaluate the influence of shared mooring line configurations.

The paper is structured as follows: Section 2 introduces the 5-MW FOWT model and proposes shared mooring layouts for the floating arrays. Section 3 describes the integrated design method and quantifies the cost-saving effect of shared mooring line configurations compared to conventional designs. The cost-driven and distance-driven mooring configurations are generated and utilized in dynamic simulations of floating turbine arrays. The simulation results and the effects of shared moorings are addressed in section 4. Finally, section 5 summarizes all findings and outlines the future work.

## 2 FOWT model and mooring configurations

The FOWT model with the coordinate system is illustrated in Figure 1. The model consists of the NREL 5-MW wind turbine (Jonkman et al., 2009) and a semi-submersible platform using a conventional mooring configuration (Robertson et al., 2014). The principal parameters of the model with the initial mooring configuration are summarized in Table 1.

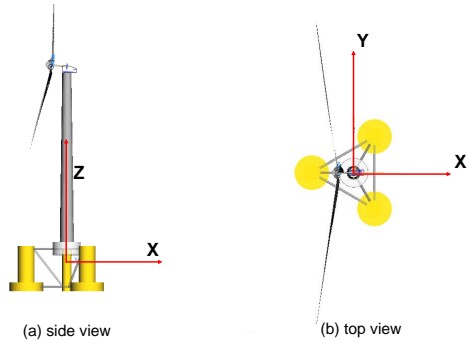

(a) side view    (b) top view

**Figure 1.** The FOWT model in (a) side view and (b) top view.

Two conventional mooring configurations, M1 and M2, are proposed for the stand-alone FOWT models, as shown in Figure 2. Each mooring line is spaced with a 120° angle between two adjacent lines. Lines L1 and L4 are parallel to the X-axis, and side lines L2 and L3 as well as L5 and L6 are symmetric about the X-axis. All the mooring lines are anchored on the seabed and their line characteristics are identical in terms of chain diameter, line length and pretension force. In configurations M1 and M2, the fairleads on the floater are positioned differently with respect to the center of the floater, resulting in a 180° deviation in the spread of three mooring lines. These two conventional mooring configurations are utilized for comparison purposes to



**Table 1.** Principal parameters of the FOWT model.

| | | |
|---|---|---|
| Wind Turbine power | MW | 5 |
| Rotor diameter | m | 126 |
| Cut-in/ Cut-out wind speed | m/s | 3 / 25 |
| Rated wind speed | m/s | 11.4 |
| Cut-in/ Rated rotor speed | rpm | 6.9 /12.1 |
| Hub height | m | 90 |
| Platform mass | kg | 1.35e7 |
| Platform displacement | m$^3$ | 1.39e4 |
| Mooring chain diameter | mm | 77 |
| Mooring line mass in air | kg/m | 113 |
| Mooring elastic stiffness | MN | 754 |
| Mooring line length | m | 836 |
| Fairlead position in Z-axis | m | -14 |

evaluate the influence of shared mooring line configurations. It is suggested to consider the effect of different fairlead positions in further research, but it is beyond the scope of this paper.

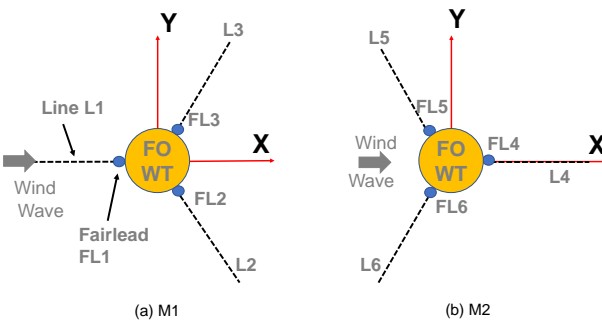

**Figure 2.** Two conventional mooring configurations (a) M1 and (b) M2.

This study utilizes three mooring lines for each FOWT and proposes two shared mooring layouts for arrays comprising of 2 and 3 FOWT models, as illustrated in Figure 3 and Figure 4 , respectively. These mooring layouts are based on studies that have presented various shared mooring designs for floating turbine arrays (Connolly and Hall, 2019; Lozon and Hall, 2023; Liang et al., 2023), as well as a handful of floating wind demonstration projects that have provided layout examples for floating wind farms.





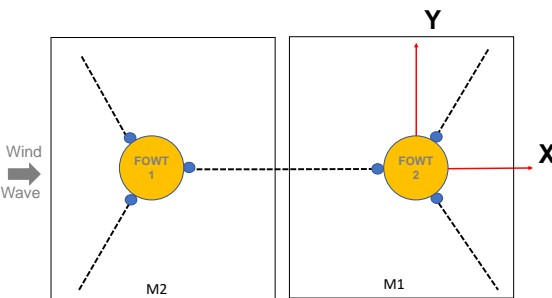

**Figure 3.** Shared mooring layout S2 for a floating array consisting of 2 FOWTs.

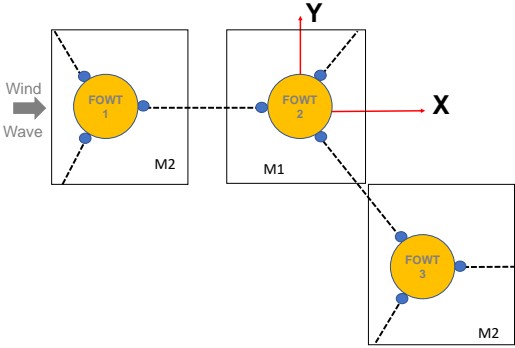

**Figure 4.** Shared mooring layout S3 for a floating array consisting of 3 FOWTs.

In the shared mooring layouts, shared mooring configurations can be viewed as an artificial superposition of conventional configurations M1 and M2. The shared mooring layout replaces specific non-shared lines in the conventional configuration with a shared line to connect two FOWT models. For instance, shared mooring layout S2 is synthetic combination of configurations M1 and M2, where fairleads FL1 and FL4 are connected by a shared line, rather than the two non-shared lines L1 and L4 in configuration M1 and M2, respectively.

Since mooring designs are tailored to specific environmental conditions of the target site, it is essential to take into account water depth information. At the site of interest in Gran Canaria, the water depth is 200 m, which aligns with typical depths for current floating wind projects, for example, the TetraSpar demonstration project (Stiesdal Offshore, 2021), but it is slightly shallower than the water depths observed in the Hywind Tampen project (Equinor, 2023). The tested turbulent wind and irregular wave conditions are presented in Table 2. The environmental conditions are based on the design basis for a reference site in Gran Canaria (Vigara et al., 2019). Two mean wind speeds $V_m$ at the hub height of the turbine are considered in the analyses, for generating the peak aerodynamic forces on the upstream and downstream turbines in two tested cases, respectively, in order to maximize floater offsets and mooring line tensions. The corresponding significant wave height $H_s$ of the irregular wave is derived from the wind-wave scatter diagram with the most probable occurrence. For the measured wave heights, the





tested wave peak period $T_p$ shows the largest probability. The study does not consider wind-wave misalignment. In the absence of public data on the turbulence intensity at target site, a Class C is assigned in the turbulent wind model, simulating moderate levels of wind turbulence (Vigara et al., 2019). It is suggested to study the effect of different wind-wave misalignment and turbulence intensity in further research, but it is beyond the scope of this paper.

**Table 2.** The tested environmental conditions.

| Test | $V_m$ [m/s] | $H_s$ [m] | $T_p$ [s] |
|------|-------------|-----------|-----------|
| T1 | 11 | 2.0 | 7 |
| T2 | 13 | 2.0 | 7 |

## 3 Mooring design and material cost evaluation

The primary goal of applying shared mooring line configurations for floating wind farms is to achieve a more cost-effective wind-farm mooring design than conventional mooring configurations used for each stand-alone FOWT. In this section, a comprehensive design method for shared mooring line configurations is presented. Also, the mooring material costs of the generated configurations are calculated to explore the cost-saving potentials of shared mooring designs. Finally, the cost-driven and distance-driven mooring designs are selected to be used for dynamic simulations.

### 3.1 The design method and workflow

The cost-based design approach that was proposed for conventional mooring configurations (Pan and Cheng, 2022) is adopted in the comprehensive design method for shared mooring configurations. Compared to the existing linearized design method, this comprehensive design approach integrates practical line characteristics, considers nonlinear mooring stiffness and multiple design constraints that align with floating wind projects and experiences from the oil & gas industry, in order to provide solid designs for both shared and non-shared mooring configurations. The workflow is depicted in Figure 5.

The design method explores a large design space with practical mooring line characteristics that are commonly utilized in current floating offshore wind projects. Design constraints consider mooring line mass, line lengths, line pretensions, lay-down length, floater horizontal excursion and a horizontal distance between two FOWTs determined by the wind-farm layout. Analytical solutions for mooring configurations are derived from catenary equations, taking into account chain elasticity. Steel chains are chosen for mooring lines, as they are the most cost-effective option for water depths less than 300 m, based on experiences from the oil and gas industry (Ma et al., 2019).

For the 5-MW FOWT model standing in the water depth of 200 m, conventional mooring configurations were found to be more advantageous than the shared mooring designs, in terms of material and installation costs(Connolly and Hall, 2019). Since it poses challenges to estimate installation costs of shared mooring designs without detailed technical specifications





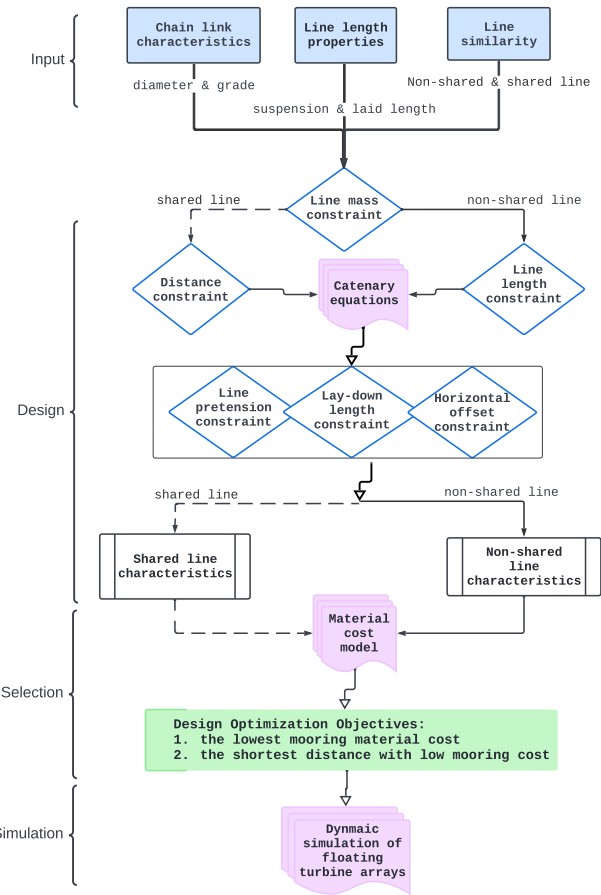

**Figure 5.** The workflow.

across different water depth at an early design phase, the focus of the cost analysis in this section is placed on mooring material costs to investigate the cost-saving potentials of shared mooring layouts proposed for floating wind turbine arrays in section 2.

In addition, the integrated design method allows for diverse design objectives. Two designs are analyzed in this paper, including the cost-driven design that achieves the lowest material costs for mooring lines and anchors, and the distance-driven design that minimizes the distance between the upstream and downstream FOWTs while maintaining low material costs. The selected configurations are then utilized in dynamic simulations of the floating array to examine the effects of shared moorings on wind power production, floater motion and mooring tension.

### 3.2 Design space of mooring configurations

The design space of mooring configurations encompasses important parameters to define mooring lines and anchors for a floating turbine array. Mooring line characteristics include chain grade ($G_c$), chain diameter ($D_c$), the length ratio ($R_L$) of the





laid length at seabed to the suspended length in waters, the length of the non-shared mooring lines ($L_{non}$), and the pretension ratio ($R_T$) of mooring pretension at fairleads over the minimum breaking load ($MBL$) of chain links.

For shared mooring lines, two additional design variables are considered: the similarity in line characteristics between
shared and non-shared lines, and the horizontal distance between the upstream turbine ($FOWT1$) and the downstream turbine ($FOWT2$). The first variable decides whether all mooring lines possess identical line characteristics, except for the line length, as the shared line length is also influenced by the horizontal distance between two FOWTs. The latter variable, expressed in terms of rotor diameter ($D_r$), is directly determined by the wind farm layout, and plays a crucial role in the performance of downstream FOWTs due to the wake effect generated by the upstream turbine. Design parameters and constraints for mooring
configurations are summarized in Table 3.

**Table 3.** Design parameters and constraints for mooring configurations.

| Variables | Values |
| --- | --- |
| Water depth | 200 m |
| Line type | Steel chain |
| Anchor type | Drag embedment anchor |
| Line arrangement | See Figure 2 to Figure 4 |
| Length ratio $R_L$ | Range [0.3 0.7] |
| Chain diameter $D_c$ | Range [60 200] mm |
| Chain grade $G_c$ | [R3, R3S, R4, R4S] |
| Characteristic similarity | [Yes, No] |
| Static mooring mass | 1.98e5 kg |
| Non-shared line length $L_{non}$ | Range [600 1000] m |
| Pretension ratio $R_T$ | Range [0.1 0.3] |
| Distance between FOWTs | Range [6 12] $D_r$ |
| Floater horizontal offset limit | [40, 60] m |
| Non-shared line lay-down length $L_{Lay}$ | above 0 m |

The simplest mooring configuration is adopted in the study, employing steel chain links for mooring lines and drag embedment anchors to minimize anchor costs (BVG Associates Ltd, 2023). Since a drag embedded anchor can only withstand horizontal forces, a portion of the mooring line must be laid on the seabed to prevent the anchor from being lifted. $R_L$ is defined as the ratio of the length laid on seabed ($L_{Lay}$) to the length suspended in water ($L_{Hang}$). $L_{Lay}$ is measured from the anchor
to the touchdown point, while $L_{Hang}$ represents the line length from the touchdown point to the fairlead. The total mooring line length is the sum of $L_{Lay}$ and $L_{Hang}$. Compared to $L_{Hang}$ derived from catenary equations, $L_{Lay}$ shows more flexibility, provided that the anchor remains unaffected by vertical tension forces. Currently, no established studies or regulations define the best practise for determining $R_L$ for FOWT models.



The range of chain diameter ($D_c$) covers various diameters utilized in floating wind projects. The range of chain grade ($G_c$)

is referenced from applications in Hywind floating wind projects (Haslum, 2019), where Hywind Demo employed R4 chain, Hywind Scotland utilized R4S studless chain, and Hywind Tampen used R3 chain in the front-end engineering design. Both $G_c$ and $D_c$ determine the minimum breaking load ($MBL$) in kN, as expressed in Equation 1.

$$MBL[kN] = f_g \times D_c^2 \times (44 - 0.08 \times D_c) \tag{1}$$

where $f_g$ is a factor dependent on $G_c$ and equals to 0.0223, 0.0249, 0.0274, and 0.0304 for R3, R3S, R4 and R4S chains,

respectively (DNV, 2018). In addition, $G_c$ has a significant impact on costs. Chains of R3, R3S, and R4 are manufactured differently from R4S chains that made by the vacuum degassed process (DNV, 2018). And this processing difference significantly influences the manufacturing cost of mooring lines. The studless chain link is employed in mooring configurations, and the mass per unit length of the mooring line in air, denoted as $Mass$ and measured in kg/m, depends on $D_c$, as shown in Equation 2 (Orcina).

$$Mass[kg/m] = 0.0199 \times D_c^2 \tag{2}$$

The elasticity of mooring lines is a crucial consideration in mooring design, and the elastic stiffness, denoted as $EA$, is calculated by incorporating the Young's modulus of the chain and the equivalent cross-sectional area of chain links. This equivalent cross-sectional area of chain links can be determined from $Mass$ and the steel density of $7850\,\mathrm{kg/m^3}$ (Orcina). $EA$ is defined in Equation 3.

$$EA[MN] = \frac{54400 \times 0.0199 \times D_c^2}{7850} \tag{3}$$

In a mooring configuration, all non-shared lines carry similar line characteristics, and shared lines possess uniform characteristics. The design variable governing the similarity of line characteristics between non-shared and shared lines determines whether all mooring lines in a shared mooring configuration exhibit identical $G_c$, $D_c$, $R_L$ and $R_T$.

### 3.3 Governing equations

The governing equations for calculating mooring tensions on segment of mooring lines are presented in Equation 4. The assumption of neglecting mooring bending and torsional stiffness is acceptable for line materials of steel chains (Faltinsen, 1993).

$$dT - \rho g A dz = [\omega \sin\phi - F(1 + T/EA)]ds,$$
$$Td\phi - \rho g A z d\phi = [\omega \cos\phi + D(1 + T/EA)]ds. \tag{4}$$





where $T$ is the mooring line tension force, $\rho g A z$ and $\omega s$ denote buoyancy and mooring line weight in water, respectively. $dz$ and $ds$ represent the vertical and total lengths of the mooring line segment, respectively, whereas $\phi$ is the spatial angle between the line segment and the horizontal plane. In addition, $F$ and $D$ are the mean hydrodynamic forces on each segment. $F$ and $D$ can be calculated by various methods, including numerical and experimental approaches. A common numerical approach involves applying the Morison equations, treating the mooring line as a slender body. Hall and Goupee (2015) introduced a lumped-mass method for modelling mooring lines model and calculated hydrodynamic forces using the Morison equations. This lumped-mass mooring line model serves as the theoretical foundation for the module of MoorDyn, and enables the prediction of dynamic tensions of mooring lines (Hall, 2015). In the static calculation, these mean hydrodynamic forces are assumed to be zero.

For mooring lines with a proportion of the length laid on the seabed, mooring tensions can be calculated from Equation 4, given knowledge of the positions of fairleads and anchors, as well as $L_{non}$, $Mass$ and $EA$ of mooring lines. These parameters can be determined by $D_c$, $R_L$, and $R_T$. Concerning shared mooring lines, current studies propose two scenarios for their spatial locations: one suspended above the seabed (Liang et al., 2020; Munir et al., 2021; Lozon and Hall, 2023), and the other partially laid on the seabed (Gözcü et al., 2022). This study considers the latter scenario, where the shared mooring line is laid on the seabed, and applies the length ratio ($R_L$) to characterize the initial proportion of the shared mooring line that is laid on the seabed in static equilibrium.

In this section, static calculations are performed to investigate the horizontal movements in surge and sway directions under a constant aerodynamic force acting on the floater. The mean aerodynamic force of 7e5 N is obtained from the steady wind simulation of the stand-alone FOWT model with the initial conventional mooring configuration from the OC4 project at the rated wind speed. This peak aerodynamic force is balanced with the mooring tension forces in the horizontal plane to derive the displacements of the floater from the catenary equations.

## 3.4 Design constraints

The first design constraint for mooring configurations is to maintain a fixed static mass of the suspended mooring lines, which ensures the hydrostatic equilibrium between the buoyancy of the platform and the total weight of the FOWT model under the consistent initial loading condition. To maintain the equilibrium is important, as different loading conditions of FOWTs can affect the dynamic performance of both the wind turbine and floater. A static mooring mass of 1.98e5 kg, calculated from the initial mooring designs in section 2, is utilized for mooring designs of the 5-MW FOWT model in static equilibrium.

This paper concentrates on mooring designs and refrains from the modification of the remaining subsystems of the 5-MW FOWT. Therefore, it assumes that no active ballast system is utilized to adjust the mass of the floater without mooring lines, in order to maintain identical hydrodynamic properties of the floater, although the static mooring mass is approximately 1.5% of the floater mass. It is also assumed that no clump weight elements are attached to the mooring lines, so the static mooring mass depends on the suspension length of the mooring line ($L_{Hang}$), the mass per unit length ($Mass$), and the number of mooring lines.





The second design constraint is related to the length of non-shared mooring lines, which ranges from 600 to 1000 m in this study, equivalent to three to five times the water depth of 200 m. This range of line lengths is derived from existing mooring designs for floating wind projects, varying from approximately 900 m for Hywind Scotland project (Equinor, 2014) to 9 times the water depth of 60 to 80 m for Kincardine Offshore Windfarm (Atkins Ltd, 2016). The length of the shared line is collectively determined by the pretension force, the distance between two FOWTs and the mooring line length ratio ($R_L$).

And the third design constraint pertains to mooring pretension force, namely the initial tension at static equilibrium without environmental loading. The range of the pretension ratio ($R_T$) is selected from 10% to 30%, aligning with common practices in the oil and gas industry (Ma et al., 2019).

The fourth design constraint focuses on the distance between the upstream and downstream FOWTs within the floating array. The distance influences the wake effect generated by the upstream turbine on the downstream turbine, which affects the wind power production of the downstream turbine. It also determines the shared mooring line length that directly impacts the mooring material cost. This study assumes similar distances between every pair of upstream and downstream turbines in the floating array. In addition, in the shared mooring layout S3 , the initial horizontal distance between $FOWT2$ and $FOWT3$ is assumed to be half of the distance between $FOWT1$ and $FOWT2$, due to the angle deviation of $120°$ between two neighboring mooring lines. The distance between two FOWTs ranges from 6 to 12 times the rotor diameter ($D_r$) of the 5-MW wind turbine, drawing reference from current layouts of offshore wind farms. For example, in the Kincardine offshore wind farm, the distances between two FOWTs vary approximately from 6 to 16 times the rotor diameter ($D_r$) of 164 m (Kincardine Offshore Windfarm Ltd, 2018), while the layout of the Hywind Scotland wind farm exhibits the distance between two FOWTs close to 9 times the rotor diameter ($D_r$) of 154 m (Jacobsen and Godvik, 2021). A handful floating wind farm projects indeed do not provide sufficient references for floating wind farm layouts. And the floating wind projects require a comprehensive assessment to determine a feasible wind-farm layout. However, this paper utilizes the layout information, instead of designing the wind-farm layout. Therefore, a board range from $6D_r$ to $12D_r$ was considered to cover various possibilities.

The horizontal offset of the floater is chosen as the fifth design constraint for mooring configurations. This offset limit is mainly governed by the design requirement of dynamic cables, which are exposed to floater movements and the environmental wave and current forces. Excessive floater displacements could increase the curvature of the cable. However, it is found that the increased curvature arising from larger floater motions is not critical for the fitness of dynamic cables, instead, the orientation of the cable relative to the floater plays a more important role (Rentschler et al., 2019). Besides, current studies lack consensus to determine the offset limit for shared mooring designs. Therefore, it is essential to maintain a reasonable floater offset in shared mooring designs, but it is not a priority to minimize offsets while ignoring other design constraints.

Two horizontal offset limits, 40 m and 60 m, are selected for the shared mooring designs of layout S2 and S3, respectively. These offset limits are based on simulations of the 2-MW FOWT with dynamic cables, which demonstrated that the cable remained in safe operation when the floater moved up to 40 m in a range of water depths from 80 m to 100 m (Taninoki et al., 2017). Given that the water depth of interest is 200 m, 60 m is assumed to be an acceptable limit for the safe operation of dynamic cables. Since the shared layout S2 has a relatively higher mooring stiffness than layout S3, due to less number of shared mooring lines, an offset limit of 40 m is used for the two-turbine array and 60 m for the three-turbine array.





The sixth design constraint, the lay-down length redundancy, places requirements on the safety of anchors. Since the mooring configuration employs the drag embedded anchor that only withstands horizontal forces, a portion of the mooring line must be laid on the seabed to prevent the anchor from being lifted. Currently, few established studies or regulations provide the range of $R_L$ for FOWT models. Therefore, a positive length limit is chosen to ensure the anchor remains unaffected by vertical tension forces.

Given that this section focuses on the design methodology rather than a specific design for shared mooring line configurations, it is recommended that future studies incorporate the up-to-date information, regarding floating wind farm layouts, site information and even installation technical specifications that affect the pretension force, in order to establish more realistic design constraints for shared mooring layouts.

### 3.5 Cost estimation for mooring configurations

This section implements the procurement cost model proposed by (Beiter et al., 2016) to calculate the minimum overall costs for mooring chains and drag embedment anchors. In this empirical cost model, the mooring line cost depends on the line length ($L$) and the minimum break load ($MBL$) of the mooring chains, while the anchor cost is expressed as a linear function of $MBL$ in Equation 5.

$$LineCost[\$] = (0.0591 \times MBL - 87.6) \times L,$$
$$AnchorCost[\$] = 10.198 \times MBL. \tag{5}$$

The comprehensive design method generated over 3000 conventional mooring configurations. The mooring material cost and the horizontal excursion for the stand-alone FOWT are compared with those of the initial mooring design from the OC4 project, as illustrated in Figure 6. Four chain grades are considered, since they determine the minimum break loads of mooring lines, which directly affects the mooring material costs. The initial design with chain grade R3 fails to meet the pretension ratio requirement. Consequently, three costs are shown for the preliminary mooring design, with chain grade R3s offering the lowest mooring material cost.

For the stand-alone FOWT model with the generated conventional mooring configuration under the peak aerodynamic force, the floater offset in surge direction ranges from 6 m to 25 m, and the mooring material cost ranges from \$6e5 to \$1.4e6. The averaged ratio of mooring line costs over anchor costs stands at 3.6, which closely aligns with the cost ratio of 3.41 in the literature (Laura and Vicente, 2014), but it exceeds the ratio of 2.89 concerning the real price in 2021 (Floating Offshore Wind Centre of Excellence, 2023). Since material costs can fluctuate across the varying floating wind projects, the deviation in the averaged cost ratio between the calculation from over 3000 configurations and the data reported from literature is acceptable, and this comparison provides reasonable evidence for validation of the cost model used in this study.

Meanwhile, for shared layouts, the design method produced more than 4000 and 6000 shared mooring configurations for the two-turbine and three-turbine arrays, respectively. Figure 7 and Figure 8 demonstrate the horizontal movement of the floater and the mooring material cost per wind turbine for the floating turbine array employing shared mooring layouts S2 and

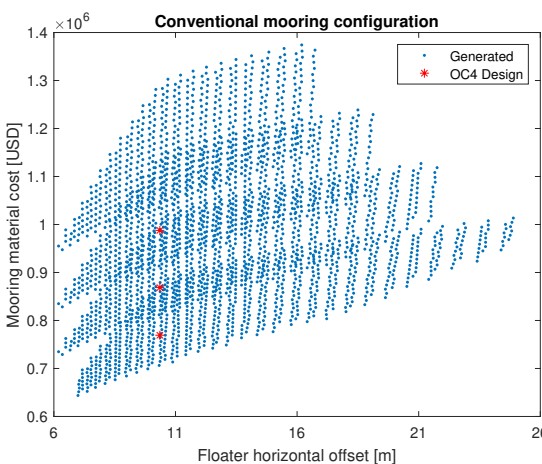

**Figure 6.** The mooring material cost and the offset for the floating wind turbine with conventional mooring configurations.

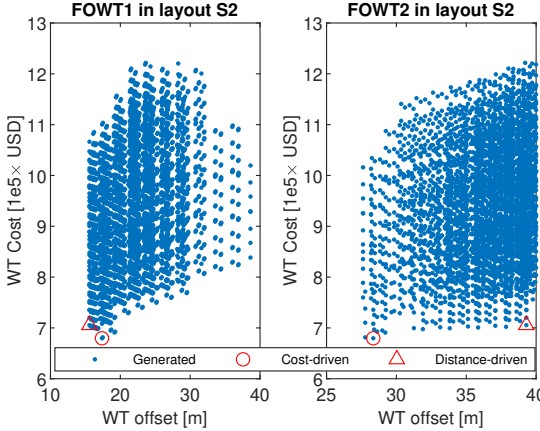

**Figure 7.** The mooring material cost and the offset for the floating wind turbine ('WT') array with the shared mooring layout S2.

S3, respectively. The cost-driven and the distance-driven mooring designs are highlighted with different symbols and will be explained with details in section 3.6.

Figure 9 illustrates the distribution of mooring material costs per wind turbine, along with their occurrences, across three groups of mooring designs: the conventional mooring configurations (M1 and M2) for the stand-alone FOWT, the shared mooring layout S2 and layout S3 for the floating turbine arrays consisting of 2 and 3 FOWT models, respectively. The bin width is 2e4$ to count the occurrence.

The material cost distribution of shared moorings is notably skewed towards lower costs compared to conventional con-
figurations. Shared mooring layout exhibits a dense cost distribution across its bins, whereas conventional moorings show a more evenly distributed pattern of mooring material costs, characterized by higher occurrences of higher costs. This compar-





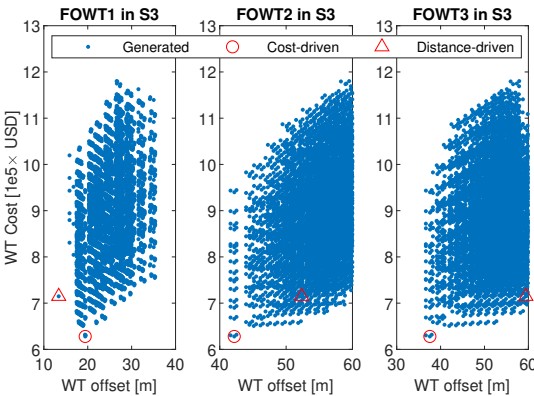

**Figure 8.** The mooring material cost and the offset for the floating wind turbine ('WT') array with the shared mooring layout S3.

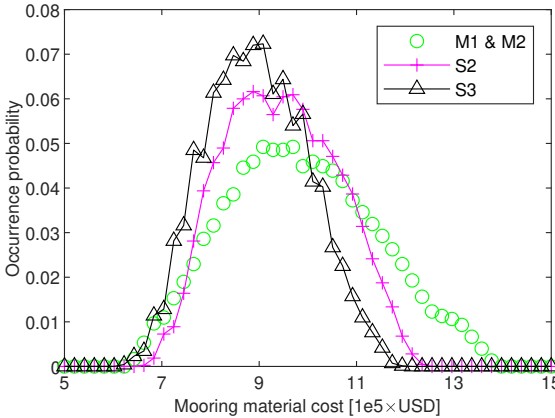

**Figure 9.** The distribution of mooring material costs per wind turbine for conventional and shared mooring designs.

ison highlights that shared mooring layouts possess the potential to achieve a more cost-effective design than conventional configurations, as they exhibit higher concentrations of lower material costs in the distribution.

Table 4 demonstrates the general trend in material cost savings associated with shared moorings. Rather than focusing on cost comparisons between specific shared and conventional mooring designs, this table presents the normalized costs across three groups of mooring designs .

The cost analysis shows that the generated shared mooring designs can reduce the mooring material costs compared to conventional mooring designs. The cost-saving benefit increases with the number of shared lines, with the shared layout S3 producing the minimum mooring material costs among the three-groups designs. Moreover, significant reductions of over 25% are observed in the standard deviation of mooring material costs, indicating that cost variations of shared mooring designs are tightly controlled compared to those of the conventional mooring configurations.





**Table 4.** The normalized mooring material costs per wind turbine for conventional and shared mooring configurations.

| Configuration | Min | Std | Mean | Max |
|---|---|---|---|---|
| M1, M2 | 1.00 | 1.00 | 1.00 | 1.00 |
| S2 | 1.06 | 0.74 | 0.96 | 0.89 |
| S3 | 0.98 | 0.68 | 0.91 | 0.86 |

This section evaluates the mooring material costs for the generated conventional and shared mooring designs, aiming to demonstrate the cost-saving potential of shared mooring designs in the early design phase. The generated shared mooring designs satisfy the six design constraints and possess the potential to provide a more cost-effective mooring solution over the
conventional mooring configuration, without considering the performance of the floating arrays under different environmental conditions.

### 3.6    Selection of shared mooring configurations

Two design objectives are proposed for the shared mooring configuration designs: a cost-driven design that focuses only on minimizing mooring material costs, and a distance-driven design that achieves the shortest distance between two connected
FOWT models while also integrating a cost-effective design. For the first objective, shared mooring configurations for layout S2 and S3 were ranked independently by their material costs, to identify the most cost-effective designs, respectively, for floating turbine arrays consisting of 2 and 3 FOWT models. Regarding the second objective, the shared configurations with the shortest distance between the upstream and downstream turbines are primarily selected. These configurations are then filtered to identify the design showing the lowest mooring material cost.
The line characteristics of selected shared mooring configurations are presented in the Table 5, including the chain diameter ($D_c$), chain grade ($G_c$), total length of mooring lines ($L_{tot}$), the length ratio ($R_L$) of the laid length at seabed to the suspended length in waters, the pretension ratio ($R_T$) of the line pretension force at fairleads over the minimum breaking load, and the distance between two FOWTs denoted by rotor diameter ($D_r$). These shared mooring configurations are intended for use in the dynamic simulations of the floating arrays to investigate the impact of shared mooring designs on the performance of turbine,
floater and mooring system.

The cost-driven and distance-driven designs yield a notable deviation exceeding over $2D_r$ in the distance between two FOWTs. This difference comes from the chain diameter being over 10% larger for the shared line in distance-driven design compared to that in cost-driven design, which results in a heavier mass density of the shared line in distance-driven design. In order to maintain a constant initial mooring suspended line mass, the adjustments are made by shortening the suspended length
in waters and the laid length on the seabed, consequently causing a smaller distance between two FOWTs.

The mooring material costs per FOWT of the cost-driven and distance-driven shared mooring configurations are normalized by those of the preliminary conventional mooring design using chain grade R3s for the stand-alone FOWT model, as summarized in Table 6. The preliminary conventional mooring design for the 5-MW FOWT model is described in section 2.



**Table 5.** The cost-driven and distance-driven shared mooring configurations.

| S2,Cost-driven | $D_c$ | $G_c$ | $L_{tot}$ | $R_L$ | $R_T$ | Distance |
|---|---|---|---|---|---|---|
| shared line | 82 mm | R4 | 1479 m | 0.3 | 0.20 | $11.7D_r$ |
| non-shared line | 76 mm | R4 | 933 m | 0.5 | 0.26 | |
| **S2,Distance-driven** | $D_c$ | $G_c$ | $L_{tot}$ | $R_L$ | $R_T$ | Distance |
| shared line | 94 mm | R4S | 1224 m | 0.4 | 0.13 | $9.5D_r$ |
| non-shared line | 76 mm | R4 | 933 m | 0.5 | 0.26 | |
| **S3,Cost-driven** | $D_c$ | $G_c$ | $L_{tot}$ | $R_L$ | $R_T$ | Distance |
| shared line | 82 mm | R4 | 1479 m | 0.3 | 0.20 | $11.7D_r$ |
| non-shared line | 74 mm | R3 | 988 m | 0.4 | 0.29 | |
| **S3,Distance-driven** | $D_c$ | $G_c$ | $L_{tot}$ | $R_L$ | $R_T$ | Distance |
| shared line | 100 mm | R3S | 1159 m | 0.5 | 0.10 | $8.9D_r$ |
| non-shared line | 76 mm | R4 | 933 m | 0.5 | 0.26 | |

Reductions in mooring material costs per FOWT are observed when utilizing both the cost-driven and distance-driven shared
mooring line configurations. As the number of shared lines increases, the cost savings become more pronounced, reaching up
to 18% for the cost-driven shared mooring designs.

**Table 6.** The normalized mooring material costs of the cost-driven and distance-driven shared mooring designs .

| Configurations | Costs | Configurations | Costs |
|---|---|---|---|
| Preliminary design | 1.00 | Preliminary design | 1.00 |
| Cost-driven,S2 | 0.88 | Distance-driven,S2 | 0.92 |
| Cost-driven,S3 | 0.82 | Distance-driven,S3 | 0.93 |

## 4 Simulation

Numerical simulations of floating arrays comprising the 5-MW FOWTs are performed in FAST.Farm, which is a mid-fidelity
tool that allows for the integrated modelling of the aero-hydro-servo-elastic analyses of FOWT models in time domain. In this
section, the simulation tool is introduced, then the static mooring stiffness is compared between shared and non-shared lines
in both the cost-driven and distance-driven shared mooring line configurations. Finally, simulation results of floating arrays





employing both shared and conventional mooring configurations are compared to evaluate the influence of shared mooring line configurations on global performances of the FOWT models.

## 4.1 Introduction to the simulation tool

To predict global performances of floating turbine arrays employing different mooring configurations under turbulent wind and irregular wave conditions, dynamic simulations are executed in FAST.Farm, which utilizes OpenFAST to solve the dynamics of each floating turbine, and incorporates the principles of the dynamic wake meandering model (DWM) to account for wake dynamics within the wind farm (Jonkman and Shaler, 2021). OpenFAST is an open-source simulation tool designed for modelling wind turbines. It provides a framework that couples various modules including Aerodyn, ElastoDyn, HydroDyn,
ServoDyn and MoorDyn, and enables the nonlinear aero-hydro-servo-elastic simulations of FOWTs in time domain (NREL, 2023).

When modelling the floating turbine array, it is crucial to take the wake effect into consideration. A wind turbine is a device that extracts kinetic energy from the wind. When the undisturbed inflow passes through the rotating blades of a turbine, the wind speed is reduced at the rotor plane and in the wake behind the rotor. The rotation of the blades adds turbulence to the
disturbed downstream airflow, leading to complex lateral and vertical expansion, as well as meandering and dissipation of the wake. Studies have revealed that the wake generated by the upstream turbines can affect the performances of downstream turbines (Wise and Bachynski, 2020; Jacobsen and Godvik, 2021), and consequently reduce the wind power production of the wind farm (Barthelmie et al., 2009; Archer et al., 2018; Ma et al., 2022).

Analytical wake models have been proposed for power prediction and the layout design of wind farms. Jensen model is the
most widely used (Archer et al., 2018), and it defines the wind speed deficit as a function of a constant wake decay factor, with the assumption of a linear wake expansion. However, the wake dynamics are not considered in the Jensen model, and require higher fidelity calculations for accurate description of wake.

In FAST.Farm, the DWM model is utilized to capture the wake-deficit evolution, the wake meandering and wake-added turbulence, which are important for the load and performance predictions (Jonkman and Shaler, 2021). Full-field turbulent
wind velocities are generated by TurbSim and incorporated into FAST.Farm as ambient inflow data. As this paper primarily focuses on assessing shared mooring configurations rather than investigating wake models for floating wind farms, consistent wake dynamic parameters are implemented in the simulations of the floating array with various mooring configurations.

## 4.2 Mooring line stiffness

For catenary lines of steel chains, the tension-displacement relationship demonstrates the mooring stiffness and characterizes
the sea-keeping capability of the mooring system. Therefore, it is essential to investigate the static mooring stiffness of both shared and non-shared lines in advance to the numerical simulations, since the restoring stiffness of the floater in surge, sway and yaw motions is provided by the mooring system. The tension-displacement relationships are compared in Figure 10 for the cost-driven and distance-driven mooring designs used for the simulation of the floating arrays of the 5-MW FOWT models.





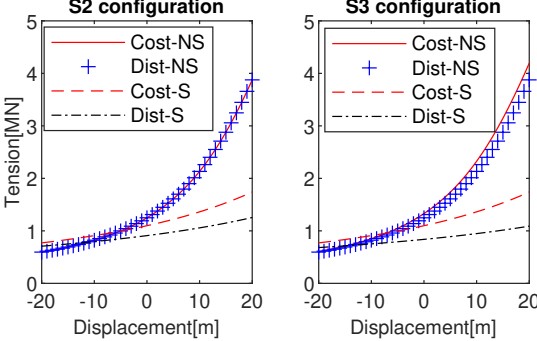

**Figure 10.** The tension-displacement relationship of shared ('S') and non-shared ('NS') mooring lines in cost-driven ('Cost') and distance-driven ('Dist') mooring designs.

The static mooring tension is calculated at fairlead by applying the catenary equations for the cost-driven and distance-driven

shared and non-shared mooring lines. A positive displacement of the shared line signifies the elongation of the horizontal distance between the upstream and downstream FOWT models, denoted as $FOWT1$ and $FOWT2$ in shared mooring layout S2, respectively. Similarly, in the conventional mooring configuration M1, a positive displacement of the non-shared line L1 indicates the expansion of the horizontal distance between the fairlead FL1 and the anchor.

At zero displacement, the mooring tension stands for the pretension force applied at fairleads. For the shared mooring layout

S2, the non-shared line experiences a pretension of 1.2 MN, while the shared line bears a pretension of 1.1 MN and 0.9 MN in the cost-driven and distance-driven designs, respectively. In addition, for the cost-driven design of shared layout S3, the pretension reaches 1.3 MN in the non-shared line and 1.1 MN in the shared line. And in the distance-driven design of layout S3, the pretension becomes 1.2 MN and 0.8 MN, in the non-shared and shared mooring lines, respectively.

The tension-displacement relationships of the non-shared lines are identical for the cost-driven and distance-driven shared

mooring designs of layout S2, but slightly different in the designs of layout S3. As for shared lines, the cost-driven design produces a higher mooring stiffness than the distance-driven design for both layouts S2 and S3. The comparisons of the tension-displacement relationships demonstrate that the mooring tension in shared lines increases at a slower rate with an increasing displacement compared to that in non-shared lines. Under similar environmental conditions, when the floating array employs a shared mooring layout, the lower and more linearized mooring stiffness of shared lines may result in larger

horizontal displacements and longer natural periods of the floater motions compared to the stand-alone FOWT model using the conventional mooring design.

### 4.3 Simulation results

Dynamic simulations of the floating array employing cost-driven and distance-driven mooring configurations are performed in FAST.Farm, incorporating two turbulent wind speeds along with corresponding irregular wave heights and peak periods. Six

random seeds are employed for each wind speed and each wave height, respectively. Each simulation spans a duration of 40





minutes, with the initial 10 minutes designated as the ramp-up period and excluded from the post-processing of results. As this paper evaluates the influence of various mooring configurations under similar environmental conditions, we presume that the number of seeds and the simulation duration are sufficient to showcase the differences among distinct mooring designs. However, it is advisable for future studies to investigate the impact of these parameters on the performance of floating arrays.

To explore the influence of shared mooring line configurations, the stand-alone FOWT models with conventional mooring configurations are used as baseline cases to compare with those FOWT modes employing shared mooring designs. The chain characteristics of non-shared lines in the cost-driven and distance-driven shared mooring configurations are applied to the mooring lines of conventional mooring configurations for each stand-alone FOWT model within the floating array. This ensures that the shared mooring lines are the only the difference between the shared and non-shared mooring configurations.

### 4.3.1    The mooring impact on power production

The mean wind power production of the FOWT models in the two-turbine and three-turbine arrays employing shared mooring configurations are compared with those using conventional mooring configurations. These power comparisons are depicted in Figure 11 and Figure 12 for the two-turbine array and three-turbine arrays, respectively.

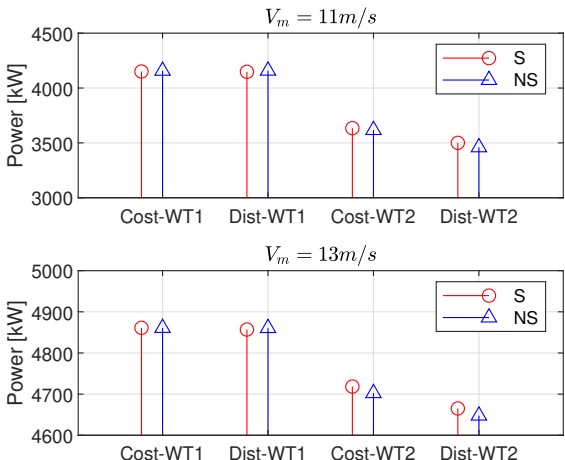

**Figure 11.** The power production of the upstream FOWT ('WT1') and downstream FOWT ('WT2') that employs shared ('S') and non-shared ('NS') mooring configurations in cost-driven ('Cost') and distance-driven ('Dist') designs.

     Under comparable upstream inflow wind conditions, the mean generated power varies slightly for a floating array using shared mooring layouts and conventional mooring configurations. The deviation between these two types of moorings is kept

within 3%. Shared mooring layouts can increase the mean wind power production of the downstream FOWT model that locates in the wake region of the upstream turbine, compared to conventional mooring designs. In the two-turbine array, at $V_m$ of 11 m/s, the upstream FOWT experiences a maximum decrease of 0.2% in mean power, while the downstream turbine exhibits a peak increase of 1.2% when applying the shared mooring layout. The increased power of the downstream turbine compensates

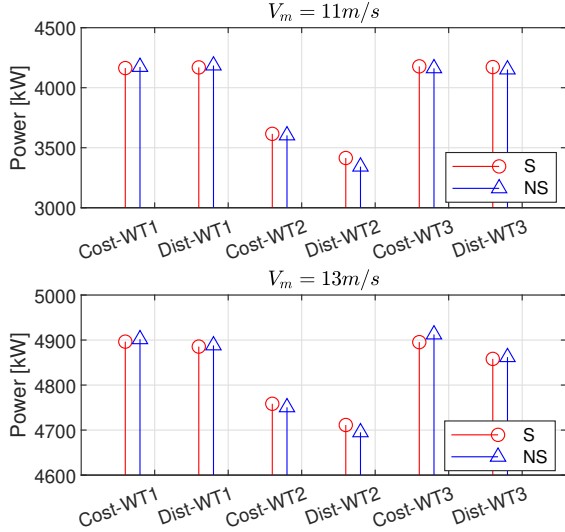

**Figure 12.** The power production of the upstream FOWT ('WT1') and downstream FOWTs ('WT2' and 'WT3') that utilize shared ('S') and non-shared ('NS') mooring configurations in cost-driven ('Cost') and distance-driven ('Dist') designs.

for the decreased power of the upstream turbine, resulting in a slight rise in the total power output of the floating array. Similarly, at $V_m$ of 11 m/s, in the distance-driven mooring designs, the shared layout S3 increases the mean power production of downstream turbine 'WT2' by 2.3%, consequently improving the total power output of the three-turbine array by 0.4%. The increase in mean power is crucial for maximizing profits of a floating array, therefore, the shared mooring line configuration demonstrates its potential to provide power enhancement solutions for the floating turbine array.

Table 7 and Table 8 present the mean wind power of each FOWT model, respectively, for the two-turbine and three-turbine floating arrays. The downstream FOWT employing the distance-driven mooring design produces lower wind power than that utilizing the cost-driven design. This is attributed to the increased distance between the upstream and downstream FOWTs in the cost-driven design, which leads to a diminished reduction in downstream wind speed. The cost-driven mooring design for the three-turbine array improves the mean wind power production of downstream turbine 'WT2', compared to the distance-driven design, with up to 5.9% rise (equivalent to 201 kW) at $V_m$ of 11 m/s.

The influence of shared mooring layouts on power generation of the downstream FOWT arises from variations in both mooring stiffness and coupling to multiple FOWTs within the floating array. Non-shared lines possess higher mooring stiffness compared to shared lines, marking the conventional configuration with the highest mooring restoring stiffness. Under similar environmental conditions, the smaller and more linearized mooring stiffness of shared mooring layouts induces larger horizontal excursions, which change the positions of the upstream and downstream FOWTs. In addition, the increased coupling to multiple downstream FOWTs in the shared mooring layouts results in different floater velocities, which impacts the relative velocity of the wind to the rotor blades. Table 9 compares the rotor-disk averaged relative wind speed for the downstream tur-





**Table 7.** The mean wind power production of the two-turbine array in the shared and non-shared mooring layouts.

| Power [kW] | $V_m = 11$ [m/s] | $V_m = 13$ [m/s] |
|---|---|---|
| Cost-WT1,S | 4151 | 4861 |
| Cost-WT1,NS | 4157 | 4861 |
| Dist-WT1,S | 4148 | 4857 |
| Dist-WT1,NS | 4158 | 4860 |
| Cost-WT2,S | 3634 | 4718 |
| Cost-WT2,NS | 3617 | 4702 |
| Dist-WT2,S | 3501 | 4665 |
| Dist-WT2,NS | 3459 | 4648 |

The upstream ('WT1') and downstream ('WT2') FOWTs employ
shared ('S') and non-shared ('NS') mooring configurations in
cost-driven ('Cost') and distance-driven ('Dist') designs.

**Table 8.** The mean wind power production of the three-turbine array in the shared and non-shared mooring layouts.

| Power [kW] | $V_m = 11$ [m/s] | $V_m = 13$ [m/s] |
|---|---|---|
| Cost-WT1,S | 4164 | 4896 |
| Cost-WT1,NS | 4173 | 4900 |
| Dist-WT1,S | 4169 | 4885 |
| Dist-WT1,NS | 4185 | 4888 |
| Cost-WT2,S | 3616 | 4758 |
| Cost-WT2,NS | 3603 | 4746 |
| Dist-WT2,S | 3415 | 4711 |
| Dist-WT2,NS | 3343 | 4695 |
| Cost-WT3,S | 4178 | 4895 |
| Cost-WT3,NS | 4162 | 4906 |
| Dist-WT3,S | 4172 | 4858 |
| Dist-WT3,NS | 4152 | 4862 |

The upstream ('WT1') and downstream ('WT2' and 'WT3')
FOWTs utilize shared ('S') and non-shared ('NS') mooring
configurations in cost-driven ('Cost') and distance-driven ('Dist')
designs.

bine 'WT2' within the two-turbine and the three-turbine arrays employing shared layouts (S2 and S3) and non-shared mooring configurations, respectively. The relative wind speed is measured normal to the disk of the rotor, taking the structural motion of the FOWT model together with the wakes effect from the upstream turbine into account (Jonkman and Shaler, 2021). The




distance-driven mooring design exhibits a larger speed deviation between the shared and non-shared mooring configurations, compared to the cost-driven design, inducing a peak difference of 1% in the mean relative wind speed for the downstream turbine 'WT2' in the three-turbine array.

**Table 9.** The mean rotor-disk averaged relative wind speed for the downstream turbine.

| Relative speed [m/s] | $V_m$= 11 [m/s] | $V_m$= 13 [m/s] |
|---|---|---|
| Cost-WT2,S2 | 10.06 | 12.07 |
| Cost-WT2,NS | 10.04 | 12.06 |
| Dist-WT2,S2 | 9.89 | 11.92 |
| Dist-WT2,NS | 9.85 | 11.89 |
| Relative speed [m/s] | $V_m$= 11 [m/s] | $V_m$= 13 [m/s] |
| Cost-WT2,S3 | 10.05 | 12.11 |
| Cost-WT2,NS | 10.04 | 12.12 |
| Dist-WT2,S3 | 9.81 | 11.90 |
| Dist-WT2,NS | 9.71 | 11.86 |

The first downstream turbine ('WT2') employs shared ('S2' for two turbines and 'S3 for three turbines) and non-shared ('NS') mooring configurations in cost-driven ('Cost') and distance-driven ('Dist') designs.

### 4.3.2 The mooring impact on floater motion

Figure 13 illustrates the mean horizontal offsets of FOWT models in the two-turbine and three-turbine arrays that utilize the shared mooring layouts S2 and S3. The mean horizontal excursions from dynamic simulations are compared with the static calculation results ('Static'), which are derived from the catenary equations and accounted for the constant aerodynamic forces acting on both the upstream and downstream FOWTs.

In general, the comparison shows that the static calculation provides a conservation prediction of the horizontal excursion, 470 compared to the numerical simulation. This offset difference arises because the steady aerodynamic forces used in the static calculation are higher than the averaged dynamic aerodynamic forces computed in simulations, and the static calculation only considers the horizontal surge and sway displacement of the platform, while ignores the yaw rotation that affects the decomposition of mooring tension in Y-axis. While the simulation considers both the structural motion of the FOWT model and the wake effect of the upstream turbine. In addition, it is observed that the static offset of the FOWTs in shared layout 475 S2 agrees better with the simulation results than that of FOWTs in layout S3. This observation indicates that the aerodynamic sensitivity of the shared mooring configuration increases with the increasing number of shared lines.

Figure 14 and Figure 15 depict the maximum amplitude of the floater's motions in six degrees of freedom for the two-turbine and three-turbine arrays, respectively. The absolute values of the motion amplitudes are compared between the shared and non-shared configurations to assess the influence of shared mooring designs. .



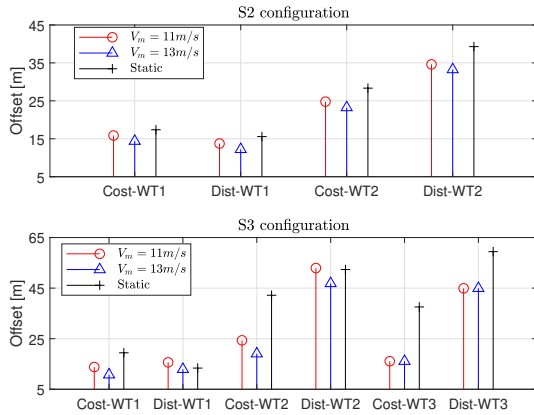

**Figure 13.** Static calculations ('Static') and simulations ('Vm') of the mean horizontal offsets for the upstream FOWT ('WT1') and downstream FOWTs ('WT2' and 'WT3') that employ cost-driven ('Cost') and distance-driven ('Dist') shared mooring designs.

Shared moorings significantly amplify surge and sway motions, compared to the conventional configuration that results in the maximum surge below 15 m and the peak sway below 1 m. In shared mooring layout S2, the shared line has a lower mooring stiffness than non-shared line L1 in conventional configuration M1 for the stand-alone FOWT model, inducing over 100% and 300% larger surge motions, respectively, for the upstream and downstream platforms under similar upstream wind conditions. As the number of shared lines increases in the shared mooring layout S3 , the horizontal mooring stiffness for the

downstream floater further diminishes, as the shared line connecting the two downstream floaters exhibits a slower tension decrease compared to the non-shared lines. Consequently, shared configurations S2 and S3 display the highest surge rise of 34 m and 51 m, respectively, at $V_m$ of 11 m/s.

     Pronounced sway differences are observed for the FOWT models in shared mooring layout S3, which is characterized by an asymmetric mooring stiffness about the Y-axis. The highest increase in the maximum sway occurs at $V_m$ of 13 m/ and

11 m/s, approaching 12 m and 39 m, respectively, in the cost-driven and distance-driven shared mooring designs. In contrast, conventional configuration M1 holds the floater within a maximum sway offset of below 1 m. The sway increase is triggered by the asymmetric mooring stiffness and the resultant tension deviation between shared and non-shared lines. The non-shared line of $FOWT3$ in layout S3, exhibits a higher mooring stiffness than the shared line connecting both $FOWT2$ and $FOWT3$. This stiffness difference creates a mooring tension deviation that pulls $FOWT3$ in a negative sway direction, leading to increased

tension in the shared line connecting both $FOWT2$ and $FOWT3$ . Consequently, this increased tension pushes $FOWT2$ in the negative sway direction as well.

     The floater's heave motion is primarily governed by the hydrostatic displacement of the platform and the total mass of the FOWT. As a result, the influence of shared moorings on floater's heave is minimal, with a maximum change below 0.2 m. Similar to the influence on sway motion, the shared mooring layout S2 , with symmetric mooring stiffness in the Y-axis, shows

little effect on roll motion. However, the asymmetric layout S3 increases the maximum rotation in roll of the downstream



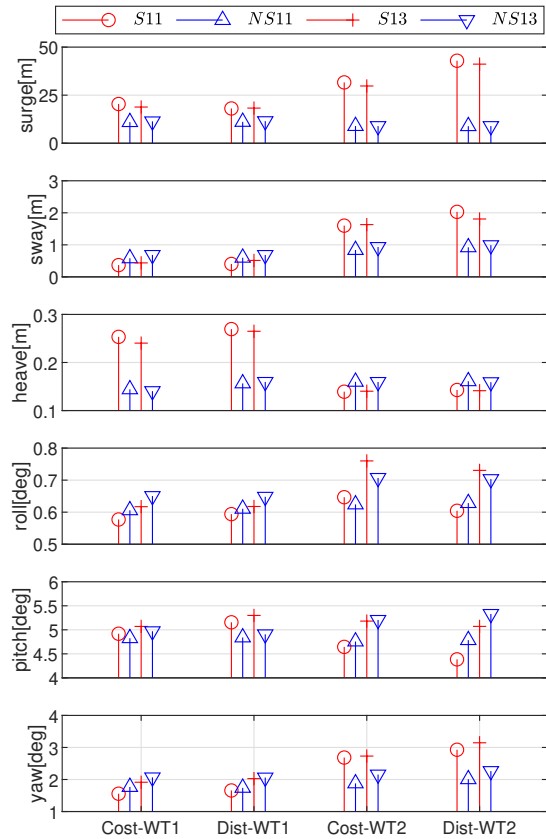

**Figure 14.** The maximum absolute amplitudes of platform motions for the upstream FOWT ('WT1') and downstream FOWTs ('WT2') that employ shared ('S') and non-shared ('NS') mooring configurations in cost-driven ('Cost') and distance-driven ('Dist') designs, at $V_m$ of 11 m/s ('11') and 13 m/s ('13').

$FOWT2$ by up to $0.2°$ and $0.4°$, respectively, in the cost-driven and distance-driven shared mooring designs, compared to the conventional mooring designs. In addition, shared moorings typically reduce floater pitch rotations of the downstream platforms. The cost-driven and distance-driven shared mooring layouts reduce the maximum pitch by up to $0.2°$ and $0.4°$, respectively.

The influence of shared mooring layouts on floater yaw rotation is similar as that on sway and roll motions, as the shared mooring layouts increase the yaw amplitudes of the downstream FOWT models. The FOWT models in the shared mooring layout S2 display the peak rise in yaw below $1°$. Shared mooring layout S3 induces a maximum increase of $3°$ and $5°$ in the maximum yaw of the downstream $FOWT3$ at $V_m$ of 13 m/s, respectively, in the cost-driven and distance-driven mooring designs, compared to the conventional mooring designs. For the downstream $FOWT2$, the increase of yaw induced by the

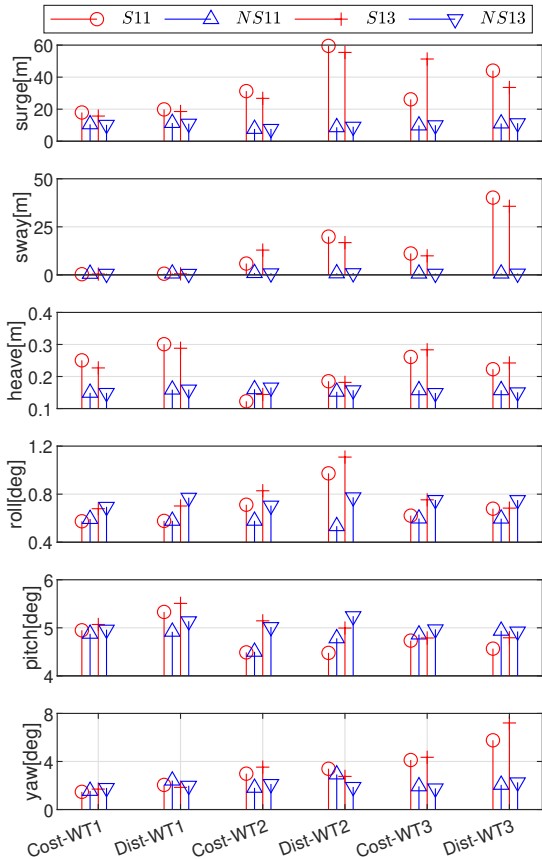

**Figure 15.** The maximum absolute amplitudes of platform motions for the upstream FOWT ('WT1') and downstream FOWTs ('WT2' and 'WT3') that employ shared ('S') and non-shared ('NS') mooring configurations in cost-driven ('Cost') and distance-driven ('Dist') designs, at $V_m$ of 11 m/s ('11') and 13 m/s ('13').

shared mooring layout S3 is below $2°$. In layout S3, the asymmetric mooring stiffness causes different mooring tensions between the shared and non-shared lines, which triggers the sway and roll motions, leading to higher tensions along the Y-axis that contributes to larger bending moments about the Z-axis.

### 4.3.3 The mooring impact on mooring tension

The maximum mooring tensions at fairleads are presented in Figure 16 and Figure 17, respectively, for the two-turbine and three-turbine arrays in shared and non-shared mooring layouts.

Under similar upstream inflow conditions, shared mooring configurations reduce the maximum tensions for the downstream FOWTs and increase the maximum tensions for the upstream FOWT, compared to conventional mooring designs. Shared

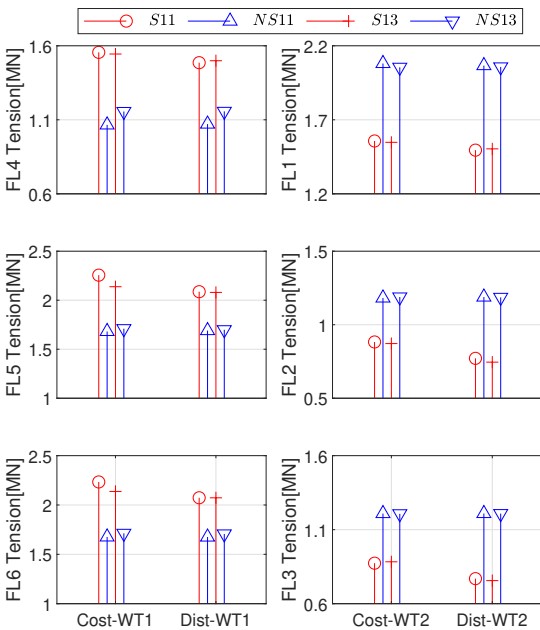

**Figure 16.** The maximum fairlead tensions for the upstream FOWT ('WT1') and downstream FOWT ('WT2') that employ shared ('S') and non-shared ('NS') mooring configurations in cost-driven ('Cost') and distance-driven ('Dist') designs, at $V_m$ of 11 m/s ('11') and 13 m/s ('13').

mooring layout S2 induces a peak deviation of 0.6 MN, resulting in a 34% rise in the maximum tension at FL5 for the upstream FOWT and a 28% drop in the maximum tension at FL1 for the downstream model. The maximum tension difference

between the shared layout S3 and conventional configurations reaches 1 MN, causing the most pronounced decrease of more than 80% in the maximum mooring tensions for the downstream $FOWT3$. The mooring tension at FL5 exhibits a peak rise of 28%, equal to 0.5 MN in the shared layout S3.

The influence of shared moorings on the redistribution of mooring tensions comes from the coupling to multiple FOWTs within the shared mooring layout, where the non-shared lines L5 and L6 that connect the upstream model are most heavily

loaded against the total environmental forces acting on the floating array. In shared mooring layouts, the reduction in mooring tensions for the downstream FOWT models arises from the significant rise of horizontal platform displacements, which notably reduces the mooring tensions in non-shared lines that connect the downstream models, compared to those in the stand-alone FOWT model using the conventional configuration.

Due to the increased mooring tension on the upstream model in shared mooring layouts, it is critical to ensure the long-term

structural integrity of the shared mooring configuration. Therefore, mooring tension fatigue damage is calculated for the three fairleads on the upstream model, applying the Rain-flow counting method and the Palmgren-Miner's rule. The slope of the S-N curve is set to 3, with an intercept of 6e10 chosen for mooring chains. The accumulated fatigue damage is averaged among the

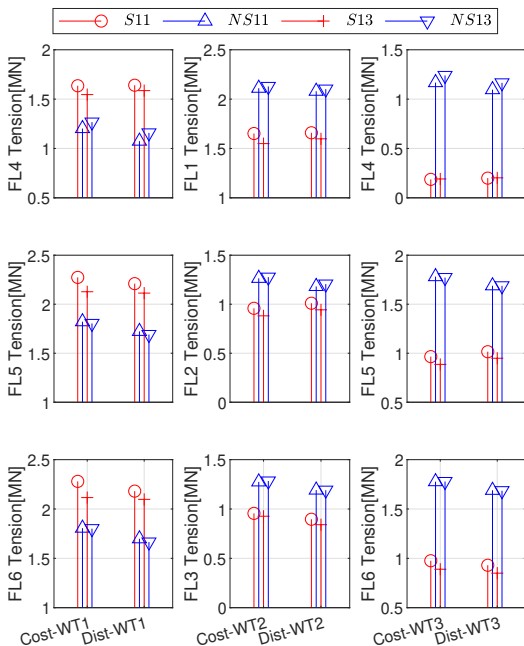

**Figure 17.** The maximum fairlead tensions for the upstream FOWT ('WT1') and downstream FOWTs ('WT2' and 'WT3') that employ shared ('S') and non-shared ('NS') mooring configurations in cost-driven ('Cost') and distance-driven ('Dist') designs, at $V_m$ of 11 m/s ('11') and 13 m/s ('13').

six seed results. The non-dimensional ratios of the mooring fatigue life for the upstream FOWT model are presented in Table 10 for the two-turbine and three-turbine floating arrays in both the cost-driven and distance-driven mooring configurations. A value
above 1 indicates that shared moorings extend the mooring fatigue life, compared to the conventional mooring configuration for the stand-alone FOWT model.

The fatigue comparison demonstrates that shared mooring configurations shorten the mooring fatigue life for the upstream FOWT model, decreasing by more than 50% for the most heavily loaded non-shared lines. Since the upstream FOWT using the conventional mooring configuration experiences a mooring fatigue life exceeding 100 years, the model employing the
shared mooring configuration still satisfies the minimum requirement of a 25-year design life. An exception is observed in the distance-driven design that elongates the mooring fatigue life at FL4 on the upstream model at $V_m$ of 13 m/s. This diverse effect is due to the fact that the tension rise at $V_m$ of 13 m/s is smaller than that that at $V_m$ of 11 m/s, and the shared mooring lines possess larger chain diameters than the non-shared lines, which collectively reduces the tension stress amplitudes and thereby results in a longer mooring fatigue life.





**Table 10.** The non-dimensional ratios of the mooring fatigue life for the upstream FOWT model.

| Two turbines | $Cost11$ | $Dist11$ | $Cost13$ | $Dist13$ |
|---|---|---|---|---|
| FL4 | 0.45 | 0.62 | 0.96 | 1.24 |
| FL5 | 0.25 | 0.32 | 0.37 | 0.40 |
| FL6 | 0.30 | 0.38 | 0.57 | 0.56 |
| Three turbines | $Cost11$ | $Dist11$ | $Cost13$ | $Dist13$ |
| FL4 | 0.69 | 0.55 | 1.17 | 1.07 |
| FL5 | 0.36 | 0.27 | 0.48 | 0.35 |
| FL6 | 0.41 | 0.36 | 0.66 | 0.48 |

The upstream model employs the cost-driven ('Cost') and distance-driven ('Dist') mooring designs at $V_m$ of 11 m/s ('11') and 13 m/s ('13').

## 5 Conclusions

By implementation of the integrated design methodology, both the conventional and shared mooring configurations are generated, and the general cost-saving trend of shared mooring designs are demonstrated. The cost analysis reveals the cost-saving advantages of shared mooring designs at a water depth of 200 m, previously considered uneconomical in the literature. The distribution of mooring material costs for over 4000 shared mooring designs is skewed towards lower costs compared to the distribution of over 3000 conventional designs. The maximum material costs of the shared mooring designs are 11% and 14% smaller than that of the conventional designs, respectively, for the two-turbine and three-turbine arrays.

Moreover, the design methodology explores the cost-driven design that achieves the lowest mooring material cost, and the distance-driven design that minimizes the wind-farm layout. Compared to the conventional mooring design from OC4 project, the cost-driven and the distance-driven shared mooring designs save 18% and 7%, respectively, in the mooring material cost for the three-turbine array. However, the quantified cost-saving benefits are constrained by reliance on the empirical cost estimation model. It is recommended that future studies incorporate the latest data from floating wind farm projects to enhance the prediction accuracy. And the cost analysis indicates the potential savings of shared moorings, as the generated mooring configurations require further studies to evaluate their performances under different environmental conditions.

Dynamic simulations demonstrate that shared mooring layouts can slightly improve the mean wind power compared to conventional mooring designs. The distance-driven shared mooring configuration induces 1% rise in the mean rotor-disk averaged relative wind speed and thus increases 2.3% in the mean power for the downstream turbine in the three-turbine array. The paper only considers two mean wind speeds to maximize the floater offset. It is suggested in the future studies to explore various wind and wave conditions to allow for extensive analysis of mooring influence on turbine performance.

In addition, the comparisons of the mean horizontal offsets indicate that the static calculation provides a conservative estimation than the simulation, due to higher aerodynamic forces and the neglect of structural motions. This also indicates that minimizing the static offset, the focus of existing linearization methods, is not a priority for shared mooring designs.

The comparison of the peak motion amplitudes shows that shared moorings amplify surge and sway motions, as well as yaw rotation of the platform, while minimally affect heave, roll and pitch motions, due to relative smaller restoring stiffness than the conventional mooring configuration. The highest surge rise reaches 34 m and 51 m, respectively, for the two-turbine and three-turbine arrays. The shared mooring layout with asymmetric mooring stiffness notably increases lateral motions in sway, roll and yaw. The highest increase in the maximum sway approaches 12 m and 39 m, respectively, in the cost-driven and distance-driven shared mooring designs.

Shared mooring layouts redistribute the mooring tensions among the connected FOWT models, resulting in the maximum tension increase of 34% for upstream model and the most pronounced tension decrease of more than 80% for the downstream model. In addition, the fatigue life comparison illustrates that shared mooring configurations shorten the mooring fatigue life of the upstream model by more than 50% for the most heavily loaded non-shared lines. These observations underscore the challenges of shared mooring designs associated with floater performances. This paper analyzes two shared mooring designs, considering the lowest mooring cost and the shortest layout distance. Future studies are recommended to optimize the shared mooring designs in order to mitigate the mooring fatigue damage and to improve the power production.

*Code and data availability.*

FAST.Farm version v3.5.0 is used in this paper and it can be downloaded from https://github.com/OpenFAST/openfast. The dataset is accessible upon the request to the authors.

*Author contributions.*

Qi Pan conceived the research question, proposed the design approach, conducted the FAST.Farm simulations, performed the data analysis and wrote the manuscript. Dexing Liu assisted in setting up the FAST.Farm model, reviewing the paper and engaging in discussions. Feng Guo supported the tuning the ROSCO controller and wake modelling. Po Wen Cheng contributed to this paper through supervision, project administration and funding acquisition.

*Competing interests.*

The authors declare that they have no competing interests.

*Acknowledgements.* The authors would like to thank Dr. Alessandro Fontanella from Politecnico di Milano for his support on wind-farm wake models and discussion on floater dynamics. Aus dem Open-Access-Publikationsfonds stehen den Forscherinnen und Forschern der Universität Stuttgart zentrale Mittel zur Finanzierung ihrer Open-Access-Veröffentlichungen zur Verfügung.



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
