# Peer review of "A Comprehensive Design Methodology of Shared Mooring Line Configurations for Assessing Mooring Costs and Performances of Floating Offshore Wind Turbines"

_Wind Energy Science, 2024_

## Referee Comment (RC1)

The paper presents a quasi-static approach to design shared mooring systems for 2 turbine and 3 turbine shared farm. Dynamic simulation for selected cases is presented using Fast.farm. The paper presents several interesting ideas. However, the conclusions arrived by comparing the shared and individual mooring is not fair as detailed in comment 10.

1. Literature survey:
    a. Line 50-54 – It is mentioned that in Liang et al (2020)'s work it was found that static mooring restoring forces are insensitive to surge and was strongly sensitive to sway. Even though the S2 layout in the current paper matches with Liang's paper, it may be a good idea to reframe this in terms of the displacements relative to the shared line headings instead because the surge and sway referred in Liang's paper is not the same as what is used in the current paper.
    b. Line 59-61 - Here a contradiction on the effect of shared line on floater motion is pointed out between studies of Munir et al. (2021) and Gözcü et al. (2022). Its unclear what is meant by "floater motions" from static tests.
2. Figure 1 and Figure 2 – Figure 1 indicates the FOWT model used in the study and I assume (since it is not specified anywhere), the same model is used for both M1 and M2 mooring configuration shown in Figure 2.
    a. For M2 configuration it is unclear how the fairleads are attached to points where there are no outer columns in the semi-sub.
    b. If M2 configuration is considered as a 180 deg. rotation of M1, meaning it requires transformation of the hydrodynamics of the semi-sub, mass properties etc. Please explain if this approach was followed as these are essential to understand the Fast.farm results presented later.
3. Figure 5 – the workflow for shared line design. Some explanation of the workflow in Figure 5 is provided in lines 223-225. But the methodology and the assumptions made are not completely clear:
    a. Please provide more clarity on what the input variables are, the bounds applied on the input variables and the constraints. In my understanding, 3000/4000/6000 designs which are generated in the study results from the permutation of 4 design variables : Length ratio, chain dia, chain grade and characteristic similarity as per Table 3 and the input block in the flow chart. If the rest are design constraints, please explain clearly how the constraints are derived from the design variables. Specifically, please explain how the following constraints are obtained from the 4 design variables using catenary equations over which a constraint is enforced:
        • Constraint 2 - Total length of anchor lines
        • Constraint 4 - distance between the turbines
    b. Is the location of the anchor fixed with respect to the platform? If so, please mention the anchor scope.
    c. In the input block, what is meant by 'Line length properties? Is a range of lengths assumed for shared and anchor line along with a $R_L$ for each? Please clarify if $R_L$ is an input variable or a design constraint as in Line 282 it is also presented as a design constraint. If it is a design constraint, please explain how this is determined from the 4 design variables using catenary equations.
4. In section 3.2 design constraints
    a. The constraint on $R_L$ already ensures that that at least 30% of suspended length is laid on the seabed for anchor lines, then what is the purpose of the additional constraint "Non-shared line lay-down length $L_{Lay} > 0$" in Table 3.

b. How do you ensure that the initial pretensions tensions as defined by the pretension ratio ($R_T$) will keep the platform in horizontal equilibrium when we consider the total force actng on the floater?

c. The term 'horizontal offset' has been used in general throughout the paper. Please define this term: are you referring to the surge displacement of the floater or the Euclidian distance which takes into account the surge and sway displacements from the initial position. Even though only 0 deg loading is considered in the paper, for the configuration S3 with 3 turbines, its essential to distinguish between the two.

d. In Line 280-281, I would think that for a given water depth, the same standard (maximum offset) will be applied for the export cables for different mooring designs. I do not follow the argument that the standard can be changed based on the realised mooring stiffness of the designed system.

e. It is unclear why strength criteria is not considered in the quasi-static mooring design workflow.

f. Line 228 Its not clear why $R_L$ is also enforced on shared line or why the shared line is designed to touch the seabed? Dragging the lines on the seabed will create large friction forces, has this been considered in the Fast.farm analysis? What is the practical relevance of this design?

5. In line 214 ρgAz is buoyancy per unit length

6. Figure 6 : Mooring material cost against offset – A general intuition is that if larger offset is permitted, the minimum mooring cost achievable would become smaller (meaning the red line in the plot will have a negative slope). Why is this trend not observed in the results?

[Figure]

7. Figure 7 and 8 – the y axes is marked as WT cost. Is this mooring cost per turbine?

8. Figure 9, Lines 322-324 – It is argued here that the skew of the shared designs indicates greater potential of shared mooring configurations to provide lower cost designs.

a. In figure 6 we clearly see 4 sets of designs corresponding to the 4 grades (I beleive). This seems to create a greater spread of the cost in the design space for individually moored case. Such an observation is not seen in the shared cases. Can you explain why this is the case. This can possibly explain the higher standard deviation and skew towards higher cost seen in the individually moored case compared to shared case.

b. If we focus on the most optimum design achievable which has the minimum cost, which is the objective of the exercise, Table 4 indicates that shared mooring configuration S2 is more expensive compared to individual mooring by around 6%. This does not exactly align with the conclusions drawn in the paper presented in the abstract or in the conclusions. Further, the savings achieved is only 2% with a 3-turbine shared mooring case over individual mooring.

9. Line 359-361 and Table 6 – Here the shared mooring configuration is shown to have a cost savings over 'preliminary' design. However, this comparison is unfair as it is not clear if the 'preliminary' design has been produced to meet the same set of constraints as that was used for producing the shared mooring designs – for example do they have the same constraint on the offset limit and pretension requirement? See comment 8b, which is a fairer comparison and shows S2 has a cost disadvantage over individually moored case.

10. Line 426-429 – It is specified that the anchor lines of the individually moored turbine is same as that used in the shared mooring implying, they are highly overdesigned as shared mooring would require the anchor lines to be stronger to account for thrust accumulation. So, any performance comparisons made between the two cannot be interpreted as a comparison of performance of an optimum shared mooring design and an optimum individually moored turbine, but a comparison between an optimum shared mooring design and a possibly overdesigned individually moored design.

11. In line 443-44 it is concluded that shared mooring shows greater potential for power production enhancement.
    a. For S2 case, it will be more interesting to look at the sway offsets seen by the turbines. See comment 10, since the conventional moored turbine is excessively stiff, I would expect it to have a lower sway displacement causing a larger power loss as it is unable to move out of the wake of the upwind turbine. I am not sure if this can be used to conclude that shared mooring can lead to higher power production.
    b. For S3, Table 8, v = 13 m/s shows that the total power production (considering 3 turbines) is in fact slightly higher for individually moored case than shared mooring cost-based design. Further, here the higher stiffness becomes an advantage for WT3 and we see a higher power production in individually moored case for both cost and distance based designs. So, I don't see any conclusive evidence of power enhancement due to shared mooring.

12. Figure 13
    a. For S3, cost driven model has a larger distance between the turbines (11.7D) as compared to distance driven model (9.5D). Therefore, I would expect a closer agreement between static results in cost driven case than in distance driven cases, as in the former case there will be a reduced wake effect. But in the figure for S3, we see a better agreement with the static results for WT2/WT3 in distance driven cases rather than cost driven case. Can you please explain this anomaly.
    b. For S3 design it appears from the comparisons presented that there is a large difference between the offsets predicted by the static tool and the actual offset seen in the dynamic simulations. If so, how effective is the design methodology proposed in the paper in identifying optimum designs in the design space?

13. Conclusions – as detailed in comment 10, the comparisons drawn between shared mooring and individual mooring can be made if the design arrived at in Table 4 M1/M2 is used ensuring they are also the most optimum.

General optional suggestions:

14. Figure 3 and Figure 4 – In the presentation of the results (for example see Figure 16 and 17) – it might be easier if the fairleads are labelled in figure 3 and figure 4 for easier interpretation of the results.

15. The term 'non-shared line' has been used throughout the text. I would recommend just using anchor line instead, if you are referring to the line connected to anchors.

---

## Author Comment (AC1)

**Replay to the first reviewer's comments**

1. Literature survey:

a. Line 50-54 – It is mentioned that in Liang et al (2020)'s work it was found that static mooring restoring forces are insensitive to surge and was strongly sensitive to sway. Even though the S2 layout in the current paper matches with Liang's paper, it may be a good idea to reframe this in terms of the displacements relative to the shared line headings instead because the surge and sway referred in Liang's paper is not the same as what is used in the current paper.

Replay: Thank you for pointing this out. We have revised the sentence and replaced 'the motions in surge and in sway directions' by 'in directions parallel and perpendicular to the heading of the shared mooring line'. Please see line 49-50.

b. Line 59-61 - Here a contradiction on the effect of shared line on floater motion is pointed out between studies of Munir et al. (2021) and Gözcü et al. (2022). Its unclear what is meant by "floater motions" from static tests.

Replay: Thanks for this comment. We have revised the sentences about the different observations from the equilibrium tests by Gözcü et al. (2022), by adding 'These equilibrium tests, which did not consider wind or wave loads, showed that shared mooring configurations significantly affect the floater displacement parallel to the heading of the shared line'. Please see line 58-59.

2. Figure 1 and Figure 2 – Figure 1 indicates the FOWT model used in the study and I assume (since it is not specified anywhere), the same model is used for both M1 and M2 mooring configuration shown in Figure 2.

Replay: Thank you for this reminder. We have added 'In this paper, the floating turbine array comprises identical turbines and platforms but utilizes different mooring configurations.' in section 2, line 105-106.

a. For M2 configuration it is unclear how the fairleads are attached to points where there are no outer columns in the semi-sub.

Replay: Thanks for mentioning this point. We have added 'The orientation of the platform employing M1 is 180 degrees opposite to that of the platform using M2, ensuring that the fairleads are always located on columns of the platform.' in section 2, line 108-109.

b. If M2 configuration is considered as a 180 deg. rotation of M1, meaning it requires transformation of the hydrodynamics of the semi-sub, mass properties etc. Please explain if this approach was followed as these are essential to understand the Fast.farm results presented later.

Replay: Thank you for this comment. We have added the hydrodynamic analysis of the platforms that consider wave directions of 0 and 180 degrees in section 4.1, see line 443-447 for details.

The public hydrodynamic files for NREL 5MW platform provided in the OC 4 project were generated for one single wave direction of 0 degree (ref1). We did not transform the hydrodynamic coefficients based on the provided hydro data. Instead, the platform is modelled in ANSYS based on the principal dimensions provided in the OC4 project, and the hydrodynamic coefficients are generated for wave directions of 0 and 180 degrees. Consequently, the hydrodynamic inputs are obtained for upstream and downstream platforms.

3. Figure 5 – the workflow for shared line design. Some explanation of the workflow in Figure 5 is provided in lines 223-225. But the methodology and the assumptions made are not completely clear:

a. Please provide more clarity on what the input variables are, the bounds applied on the input variables and the constraints. In my understanding, 3000/4000/6000 designs which are generated in the study results from the permutation of 4 design variables : Length ratio, chain dia, chain grade and characteristic similarity as per Table 3 and the input block in the flow chart. If the rest are design constraints, please explain clearly how the constraints are derived from the design variables.

Specifically, please explain how the following constraints are obtained from the 4 design variables using catenary equations over which a constraint is enforced:
• Constraint 2 - Total length of anchor lines
• Constraint 4 - distance between the turbines

Replay: Thank for this comment. We have revised section 3.1 regarding the workflow. Please see line 168- 171 for the description of the input design variables. The flowchart in Figure 5 has been updated to separately include static calculations without and with wind force, reflecting the design process.

In addition, section 3.4 has been revised to detail the design process and the functioning of the design constraints. Regarding the use of catenary equations: for the constraints 1 & 2 (line mass and line length), catenary equations are not used, while for constraints 3 to 6 (pretension, FOWT distance, offset, and lay-down length redundancy), elastic catenary equations are applied to check the mooring designs under no wind and peak wind force conditions, respectively. Please see line 291-295 for details.

How total length is obtained: Please see line 270-275 and 284-287 for details.
1. Given the chain diameter, the static mooring line mass is calculated as the product of mass density per unit length and the suspended length in water. The mass density of a mooring line in water is equal to the mass density in air minus the displaced mass of water resulting from the equivalent cross-sectional area of the chain links. Constraint 1 filters the mooring designs and determines the suspended length for each chain diameter.
2. Next, the initial lay-down length is determined based on the length ratio, which ranges from 0.3 to 0.7. Consequently, the total length of a mooring line is obtained by summing the hanging length and the lay-down length.

How FOWT distance is obtained: Please see line 300-304 for details.
1. For shared mooring designs, this distance depends on the fairlead radius, the horizontal distance from fairlead to the touch-down point and the laid length on seabed.
2. Given the chain diameter and the suspended length, elastic catenary equations are used to calculate the horizontal distance from the fairlead to the touch-down point, based on the mass density in water and the vertical distance from the fairlead to the seabed.

b. Is the location of the anchor fixed with respect to the platform? If so, please mention the anchor scope.

Replay: The anchor position is not fixed, but the fairlead radius remains consistent with OC4 projects. We have added this radius in Table 1.

c. In the input block, what is meant by 'Line length properties? Is a range of lengths assumed for shared and anchor line along with a RL for each? Please clarify if RL is an input variable or a design constraint as in Line 282 it is also presented as a design constraint. If it is a design constraint, please explain how this is determined from the 4 design variables using catenary equations.

Replay: In the input block of Figure 5, the line length properties include the mooring line suspension length in water and the laid length on seabed.

The length properties can be a range of lengths for shared and anchor lines, but this paper considers a constant mooring mass in the static condition. Therefore, for each chain diameter, the mass density of the mooring line is calculated, and the initial suspended length is determined by the static mooring mass. The initial laid length is then calculated from RL. The length ratio (RL) is an input design variable for calculations of the mooring line total lengths. Please see section 3.2, line 205-210 for details.

In line 282 of the draft, the lay-down length redundancy is not the initial laid length under the static conditions without wind and wave forces. Instead, this length redundancy is the limit of laid length under the peak thrust condition when the floater experiences the maximum offset. We have revised Table 3 and Figure 5 to clarify this distinction.

4. In section 3.2 design constraints
a. The constraint on RL already ensures that that at least 30% of suspended length is laid on the seabed for anchor lines, then what is the purpose of the additional constraint "Non-shared line lay-down length LLay > 0 " in Table 3.

Replay: Thanks for mentioning this point. The length ratio (RL) is an input design variable for the mooring line length, used to calculate the laid length from the suspended length in static conditions without wind forces, in order to determine the total length.

The sixth constraint, however, addresses the redundancy of the lay-down length under the peak wind force. This ensures the seabed contact for the safety of anchors and for the validity of using catenary equations for shared lines. Please see line 335-340. We have revised the Table 3 and Figure 5 to clarify this distinction.

b. How do you ensure that the initial pretensions tensions as defined by the pretension ratio (RT) will keep the platform in horizontal equilibrium when we consider the total force actng on the floater?

Replay: The pretension ratio can't guarantee the initial horizontal equilibrium. This design constraint (RT) considers the installation practices from oil and gas industry, rather accounting for the horizontal force equilibrium.

c. The term 'horizontal offset' has been used in general throughout the paper. Please define this term: are you referring to the surge displacement of the floater or the Euclidian distance which takes into account the surge and sway displacements from the initial position. Even though only 0 deg loading is considered in the paper, for the configuration S3 with 3 turbines, its essential to distinguish between the two.

Replay: Thank for this point. It is the Euclidean distance that accounts for both the surge and sway excursion from the initial position. We have added explanations in section 3.4, line 317.

d. In Line 280-281, I would think that for a given water depth, the same standard (maximum offset) will be applied for the export cables for different mooring designs. I do not follow the argument that the standard can be changed based on the realised mooring stiffness of the designed system.

Replay: Thanks for this comment. It is common practice to apply the same standard, such as offset limit, to evaluate different mooring configurations for stand-alone floating turbines. When compare shared mooring configurations for the same floating arrays, it is more reasonable to use a constant offset limit for both S2 and S3. However, the goal of this paper is to design S2 for the two-turbine array and S3 for the three-turbine array, respectively. For each floating array, the same offset limit is employed.

As for the use of 40 m for S2 and 60 m for S3 aims to establish reasonable offset limits for different floating arrays, since both 40 m and 60 m are acceptable at the water depth of 200 m. A similar study on shared mooring design applies 60 m for the static test and 72 m for the dynamic simulation (ref2). This illustrates the application of reasonable offset limits for different analysis purposes. Please see line 325-334 for more details.

e. It is unclear why strength criteria is not considered in the quasi-static mooring design workflow.

Replay: Thanks for pointing this out. It is important to consider strength in the design process. The quasi-static workflow did not include the line strength as an additional design constraint, because the design constraints on pretension ratio and lay-down length redundancy already limit the maximum tension within the minimum breaking load. Here are how these constraints work:

The total tension force $T$ on the fairlead can be expressed by:
$$T = \sqrt{Tz^2 + Th^2} \qquad \text{Eq1.}$$

Where the vertical tension component $Tz = ls \times \omega$ and the horizontal component $Th = la \times \omega$. The line weight density $\omega$ is function of mass density and is determined by chain diameter, so for each mooring line, $\omega$ remains constant.

Based on the spatial position relationship for catenary lines and assuming no elasticity for simplification, the relationship between the suspension length $ls$ and the length for horizontal tension component $la$ is denoted as:

$$ls^2 = h^2 + 2h \times la \qquad \text{Eq2. (ref3)}$$

Where $h$ is the vertical distance from anchor to fairlead. By combination of Eq1 and Eq2, we can see that
$$T = (ls^2 + h^2) \times \omega / (2 \times h) \qquad \text{Eq3.}$$

For quasi-static design process, both $h$ and $\omega$ keep constant, so the change of $T$ depends on the change of $ls$.

- The design constraint on the lay-down length redundancy sets the boundary for $ls$, ensuring the maximum $ls$ within the total line length $l_{total}$.

This paper considers zero as the lay-down length limit under the peak thrust force, therefore, maximum suspension length $l_{smax}$ is $l_{total}$.
$$l_{smax} = l_{total} = l_{s0} \times (1 + R_L) \qquad \text{Eq4.}$$

Where $l_{s0}$ is the initial suspension length that corresponds to the pretension $T_0$, and $R_L$ is the length ratio in the static condition without wind or wave forces.

For $l_{smax}$, applying Eq3 to get the maximum tension $T_{max}$, so that

$$\frac{T_{max}}{T_0} = \frac{l_{smax}^2 + h^2}{l_{s0}^2 + h^2} = \frac{l_{s0}^2 \times (1 + R_L)^2 + h^2}{l_{s0}^2 + h^2} < (1 + R_L)^2$$

when $R_L = [0.3\ 0.7], \frac{T_{max}}{T_0} < 2.89$

- The design constraint of pretension ratio limits the pretension $T_0$ over the minimum breaking load $MBL$ to the range of 0.1 to 0.3.

as $T_0 = [0.1\ 0.3] \times MBL$, then $T_{max} < 0.3 \times 2.89 \times MBL$
so $\frac{T_{max}}{MBL} < 0.87$

Therefore, $T_{max}$ is below $MBL$ and the tensile strength is secured by the existing design constraints.

f. Line 228 Its not clear why RL is also enforced on shared line or why the shared line is designed to touch the seabed? Dragging the lines on the seabed will create large friction forces, has this been considered in the Fast.farm analysis? What is the practical relevance of this design?

Replay: thank you for mentioning these points. For shared lines without anchors, the motivation of applying length ratio is to ensure the adequate contact on seabed, so that it is valid to apply the elastic catenary equations. For hanging lines without seabed contact, using elastic catenary equation is invalid, in such case, the hanging cable equation proposed by Irvine is more appropriate (ref4). Please see line 338-341 for details.

As for the seabed friction force, it is simulated by the module Moordyn (version V2) in Fast.Farm, by applying 1.0 and 0.69 for the transverse and axial friction coefficient. We have added this explanation in section 4.1, see line 448-449.

The reason for using a partially lay-down shared line on seabed is its simplicity compared to the hanging lines. A risk associated with hanging lines is the potential conflict with passing ships or fishing nets. To avoid such conflicts, additional devices to restrict line movement within allowable limits are required, such as clump weights and buoys, which increases the complexity of shared mooring designs. We have added the explanations in section 3.3, see line 252-256.

5. In line 214 ρgAz is buoyancy per unit length
Replay: dz is the vertical distance of a line segment, then ρgAdz represents the increased buoyance of this line segment. Therefore, ρgAz is the buoyance for the line. See the figure below (ref3).

[Figure]

Fig. 8.2. Forces acting on an element of an anchor line.

Figure: force on the line segment (ref3)

6. Figure 6 : Mooring material cost against offset – A general intuition is that if larger offset is permitted, the minimum mooring cost achievable would become smaller (meaning the red line in the plot will have a negative slope). Why is this trend not observed in the results?

Replay: thank you for this comment. Regarding the comment on material cost versus offset, for a constant mooring line length and a constant pretension force, it is true that a larger allowable offset lowers the mooring cost. This is achieved by using a smaller chain diameter or/and lower grade, since the requirement for mooring stiffness is less strict. However, when mooring line lengths and pretensions vary, this statement does not hold, because the material cost is determined by length, chain grade and diameter.

Due to design constraint 1, the initial suspension length $l_{s0}$ increases as chain diameter decreases. As a result, the total length is not constant across all configurations, which causes the material cost no longer monotonically decreasing with smaller chain diameter or lower grade. Instead, the cost is also influenced by varying total lengths. In addition, pretensions vary across all configurations. Lower mooring stiffness is not only due to smaller chain diameter, but also from smaller mooring tension. Therefore, because of these varying line lengths and diverse pretensions, we did not observe a negative slope in Figure6.

7. Figure 7 and 8 – the y axes is marked as WT cost. Is this mooring cost per turbine?

Replay: thanks for this reminder. Yes, it is the material cost per wind turbine. We have revised the label in Figure 7 and 8.

8. Figure 9, Lines 322-324 – It is argued here that the skew of the shared designs indicates greater potential of shared mooring configurations to provide lower cost designs.
a. In figure 6 we clearly see 4 sets of designs corresponding to the 4 grades (I beleive). This seems to create a greater spread of the cost in the design space for individually moored case. Such an observation is not seen in the shared cases. Can you explain why this is the case. This can possibly explain the higher standard deviation and skew towards higher cost seen in the individually moored case compared to shared case.

Replay: thank you for this comment. The chain grade affects the MBL for the mooring line and thus influences the mooring material cost. For the shared configuration case, the impact of chain grade on material costs is less evident as it is for the conventional mooring case.

The first reason is that within the shared mooring configuration, the shared and non-shared lines can possess different chain grades and diameters. While in conventional configurations, all lines have identical chain properties. The second reason is that for shared configuration, the contribution of MBL in the total material cost is less significant, compared to conventional configurations that included extra anchor costs equal to 10.198 x MBL.

b. If we focus on the most optimum design achievable which has the minimum cost, which is the objective of the exercise, Table 4 indicates that shared mooring configuration S2 is more expensive compared to individual mooring by around 6%. This does not exactly align with the conclusions drawn in the paper presented in the abstract or in the conclusions. Further, the savings achieved is only 2% with a 3-turbine shared mooring case over individual mooring.

Replay: thanks for these comments. Two design targets, including lowest cost and shortest distance designs, are used for dynamic simulations to investigate the influence of shared mooring line. Table 4 shows that the lowest cost increases by 6% for S2 and decreases by 2% for S3. This occurs because the design constraint 'FOWT distance' imposes additional limits on the length of shared lines and affects the material cost. We had added the explanation in section 3.5, line 388-390.

Concerning the conclusion, it was stated that 'The maximum material costs of the shared mooring designs are 11% and 14% smaller than that of the conventional designs, respectively, for the two-turbine and three-turbine arrays.' The reduction in the maximum cost indicates that shared moorings possess the potential to lower the most expensive scenarios, which mitigates the financial risk and makes the shared mooring design more attractive for stakeholders who are concerned about the worst-case scenarios. We had added these explanations in line 393-395.

In addition, the overall trend indicated evident cost savings for shared moorings, which exhibit a concentration in the lower cost range (see Figure 9 for the cost distribution of the three groups). The denser distribution in the lower cost range suggests that shared mooring designs can be more predictably cost-effective, compared to the conventional configurations.

9. Line 359-361 and Table 6 – Here the shared mooring configuration is shown to have a cost savings over 'preliminary' design. However, this comparison is unfair as it is not clear if the 'preliminary' design has been produced to meet the same set of constraints as that was used for producing the shared mooring designs – for example do they have the same constraint on the offset limit and pretension requirement? See comment 8b, which is a fairer comparison and shows S2 has a cost disadvantage over individually moored case.

Replay: thanks for these comments. The preliminary mooring design from the OC4 project meets the design constraints, as described in section 3.5 line 354-360. Four chain grades were used, but grade R3 failed to meet the requirement of pretension ratio. For conventional mooring configurations, the offsets under the peak thrust are within 30 m, and the preliminary design produced the offset of less than 11 m under the thrust wind force (see Figure 6).

The lowest material cost of three preliminary mooring designs is used as the baseline case for comparison with the two selected shared mooring designs. This is due to the fact that the analyzed FOWT model is based on the models from the OC4 project, and the static mooring line mass is derived from the preliminary mooring design.

10. Line 426-429 – It is specified that the anchor lines of the individually moored turbine is same as that used in the shared mooring implying, they are highly overdesigned as shared mooring would require the anchor lines to be stronger to account for thrust accumulation. So, any performance comparisons made between the two cannot be interpreted as a comparison of performance of an optimum shared mooring design and an optimum individually moored turbine, but a comparison between an optimum shared mooring design and a possibly overdesigned individually moored design.

Replay: thanks for this comment. The purpose of comparing the dynamic performance of floating arrays is not to compare an optimal shared design (lowest cost) with an optimal conventional mooring design (lowest cost). Two design targets are selected for shared mooring configurations used in dynamic simulations, in order to investigate the influence of shared mooring lines on the performance of floating turbines. These two design goals are not set for the conventional designs. No optimal conventional design with the lowest costs or the shortest distance was generated for the stand-alone FOWTs.

The non-shared lines from the selected shared mooring designs were applied to the stand-alone turbines in the dynamic simulations. This can ensure that the shared mooring lines are the only difference between shared and conventional configurations for the same floating array. Therefore, this paper does not address the optimized designs of shared versus conventional mooring configurations.

11. In line 443-44 it is concluded that shared mooring shows greater potential for power production enhancement.
a. For S2 case, it will be more interesting to look at the sway offsets seen by the turbines. See comment 10, since the conventional moored turbine is excessively stiff, I would expect it to have a lower sway displacement causing a larger power loss as it is unable to move out of the wake of the upwind turbine. I am not sure if this can be used to conclude that shared mooring can lead to higher power production.

Replay: thanks for this point. Figure 14 compares the maximum motion of the floaters with shared and conventional mooring configurations. The downstream WT2 with the conventional configuration has the maximum sway of 1 m. While with S2, the maximum sway reaches 2 m. And S2 increases the yaw motion of WT2 increases by less than 1 degree.

The influence of the shared mooring line on floater motion collectively affects power production. Under comparable upstream inflow conditions, S2 slightly increases the rotor-disk averaged relative wind speed of the downstream turbine WT2 (see Table 9). As a result, S2 increases the mean power of WT2 by up to 1.2% at V = 11 m/s, compared to the stand-alone turbine.

b. For S3, Table 8, v = 13 m/s shows that the total power production (considering 3 turbines) is in fact slightly higher for individually moored case than shared mooring cost-based design. Further, here the higher stiffness becomes an advantage for WT3 and we see a higher power production in individually moored case for both cost and distance based designs. So, I don't see any conclusive evidence of power enhancement due to shared mooring.

Replay: thanks for these comments. At V = 13 m/s, the cost-driven and distance-driven shared configurations S3 result in a decrease of 3 kW and an increase of 9 kW in the total power production, respectively, compared to the conventional design. While at V = 11 m/s, S3 increase the total power by more than 20 kW. Overall, the shared mooring configuration results in a net increase in total power production across both wind speed conditions. We have revised the description in section 4.3.1, line 537-540.

Similarly, for the downstream WT3, S3 reduces the mean power by less than 10 kW at V = 13 m/s, but increases it by more than 15 kW at V = 11 m/s. Overall, S3 results in a net increase in the power output of WT3.

12. Figure 13
a. For S3, cost driven model has a larger distance between the turbines (11.7D) as compared to distance driven model (9.5D). Therefore, I would expect a closer agreement between static results in cost driven case than in distance driven cases, as in the former case there will be a reduced wake effect. But in the figure for S3, we see a better agreement with the static results for WT2/WT3 in distance driven cases rather than cost driven case. Can you please explain this anomaly.

Replay: thank you for pointing this out. The smaller FOWT distance, the strong wake effect on the downstream turbine WT2. The cost-driven design has a larger distance, so the wake deficit becomes smaller. Table 9 shows that the cost-driven design S3 gives 0.2~0.3 m/s higher speed than the distance-driven one for WT2. This velocity rise induces less than 3% difference in the aerodynamic force. Therefore, the distance difference between two designs has a limited effect on the offset difference between static and dynamic results.

Instead, the neglect of structural rotations has a more evident effect. The static calculations only consider surge and sway, and ignore the rotation of the floater. The mooring force is calculated based on the fairlead and anchor positions. The pitch affects the fairlead positions in X and Z-axis, and the yaw affects the fairlead positions in X and Y-axis. Also, the mooring tension is balanced with the peak thrust but the yaw influences the mooring force decomposition in the horizontal plane. Therefore, the neglect of rotation causes deficit in the fairlead position and the mooring tensions between static and dynamic results.

The higher the mooring stiffness, the more sensitive the mooring tension is to the fairlead position. Figure 10 shows that the shared line in cost-driven design has a higher mooring stiffness than that in the distance-driven one. As a result, the neglect of rotation has a stronger impact on the cost-driven design compared to the distance-driven design, and causes a larger offset difference between static and dynamic results. We have added these explanations in section 4.3.2, line 555-561.

b. For S3 design it appears from the comparisons presented that there is a large difference between the offsets predicted by the static tool and the actual offset seen in the dynamic simulations. If so, how effective is the design methodology proposed in the paper in identifying optimum designs in the design space?

Replay: thanks for this comment. This paper proposes two design goals including lowest cost and shortest distance between FOWTs, rather than minimization of the static offset, which is the concern of existing linearized design methods. The static calculation predicts a larger offset than the dynamic simulation. Given a constant offset limit, the dynamic simulation can generate more viable designs than the static approach.

The cost-driven design locates in the small-offset region, see Figure 7-8. It is not affected by the offset prediction bias between static and dynamic results, because the general trend indicates a larger offset gives higher WT material cost (see Figure 7-8). Designs that satisfy the dynamic simulation but fails the static approach tend to be more expensive. If the optimization objective is lowest cost, then the proposed method is effective to generate the optimal mooring design, without running the dynamic simulation.

13. Conclusions – as detailed in comment 10, the comparisons drawn between shared mooring and individual mooring can be made if the design arrived at in Table 4 M1/M2 is used ensuring they are also the most optimum.

Replay: thank you for this comment. Two shared mooring configurations are selected for dynamic simulations, in order to investigate the influence of shared mooring lines on the performance of floating turbines. The goal of dynamic simulation is not to compare an optimal shared design (lowest cost) with an optimal conventional mooring design (lowest cost).

In the dynamic simulations, we utilize similar chain properties for non-shared lines in the shared and conventional mooring layout to exclude the influence of non-shared lines, in order to ensure shared mooring lines being the only difference between shared mooring layout and conventional configurations for the same floating array.

General optional suggestions:
14. Figure 3 and Figure 4 – In the presentation of the results (for example see Figure 16 and 17) –it might be easier if the fairleads are labelled in figure 3 and figure 4 for easier interpretation of the results.

Replay: thanks for this point, we have added the names of fairleads, see Figure 3 and Figure 4.

15. The term 'non-shared line' has been used throughout the text. I would recommend just using anchor line instead, if you are referring to the line connected to anchors.

Replay: thanks for this suggestion. 'anchor line' is commonly used in the literature for mooring lines, 'non-shared line' provides a clearer distinction between the two types of mooring lines (shared v.s non-shared).

**Reference:**

[1] NREL. https://github.com/OpenFAST/r-test/tree/main/glue-codes/openfast/5MW_Baseline/HydroData
[2] Wilson, Samuel, et al. "Linearized modeling and optimization of shared mooring systems." Ocean Engineering 241 (2021): 110009.
[3] Faltinsen, Odd. *Sea loads on ships and offshore structures*. Vol. 1. Cambridge university press, 1993.
[4] Liang, Guodong, Karl Merz, and Zhiyu Jiang. "Modeling of a shared mooring system for a dual-spar configuration." *International conference on offshore mechanics and arctic engineering*. Vol. 84416. American Society of Mechanical Engineers, 2020.

**Replay to the second reviewer's comments**

This manuscript presents a preliminary design approach for shared mooring systems focusing on static responses and material costs. Results of dynamic analysis are presented and discussed for selected design candidates. However, whether the selected design sea states are representative for mooring design at the considered offshore site is questionable. Additional information is needed.

Replay: thank you for this comment. This paper addresses the small-offset limitation existed in the current linearized design methods. To estimate the maximum offset, the peak thrust force is applied in the static calculations, similar to the approach in studies (ref1-ref2). The resulting static offset is then compared with the dynamic simulation result. The dynamic simulations are performed to investigate the influence of shared mooring configurations, considering the normal operational scenarios.

To explore the representative sea states for floater movements, we refer to the dynamic simulations of the 15-MW semi-submersible floating turbine at the same site. The report (ref3) indicates that, among all tested wind speeds, the rated wind speed is the most critical for floater movements. At the rated wind speed, DLC1.3 (extreme turbulence wind + stochastic wave Hs = 2m, similar to the test case in this paper) induces larger motions in the maximum surge and pitch than DLC1.6 (normal turbulence wind+ extreme stochastic wave Hs = 5 m). In addition, since mooring tensions are determined by the floater motions, DLC1.3 consequently generates the largest tensions in mooring lines. Therefore, the selected environmental conditions in the paper can represent the most critical sea states for the floating array at the site of interest. We have added these explanations in section 2, line 137-144.

Some additional comments are as follows:
1. In Sec.2, how are the two wind speeds selected? Do they refer to certain return period? Please provide more information.

Replay: thank you for this point. The two mean wind speeds (11 m/s and 13 m/s) are selected to ensure both the upstream and downstream turbines can experience the maximum aerodynamic forces in the two test cases, respectively, as the rated wind speed for the analyzed turbine is 11.4 m/s. The dynamic simulations of the 15-MW semi-submersible floating turbine at the same site (ref3) indicates that, among all tested wind speeds, the rated wind speed is the most critical for floater movements.

This paper does not consider the 50-yr extreme wind condition for a parked turbine, because it focuses on the normal operation scenario of the wind turbine. DLC1.3 refers to the design situation of power production, and is similar to the test in this paper. In addition, the environmental conditions are based on a reference site in Gran Canaria (ref4). And the dynamic simulations of the 15-MW semi-submersible floating turbine at the same site (ref3) shows that DLC1.3 (extreme turbulence wind + stochastic wave Hs = 2m, similar as the test case in the paper) results in higher floater motions than DLC6.1 (parked turbine scenario, extreme 50-yr wind + wave Hs = 5 m). Therefore, the two test cases are the most critical for the floater movements and mooring tensions. We have added these explanations in section 2, see 137-144.

2. How will current affect the design results?

Replay: thank you for mentioning this point. We have added explanations in section 3.3, see line 244-247 for static calculation, and in section 2, see line 149-155 for dynamic simulation, respectively.

The elastic catenary equation neglects the current force, because this paper considers the normal operation scenario of the turbine, where the influence of current force is limited on mooring tension. If other scenario is considered and current force is included, the catenary equation becomes non-linear, making it generally impossible to derive an explicit solution in static calculations (ref5).

For the dynamic simulation, this paper consideres the normal operational scenario of the turbine and neglects the current in the hydrodynamic calculation. This assumption is acceptable, because the current speed at the

sea surface is primarily driven by strong wind, and the 50-yr extreme wind speed of 19 m/s at 10 m reference height induces a current speed of 0.57 m/s at the sea surface of the site (ref4). Therefore, it is expected the influence of the current on the floater movement and on the mooring line tension is limited for the normal operational scenario of the turbine, since the mean wind speeds at hub height are 11 m/s and 13 m/s, much smaller than the extreme wind speed of 19 m/s at 10 m height.

3.  In Sec.3.5, how the FOWT offsets are obtained for different mooring configurations in the design space? Please add more details.

Replay: thank you for this point. We have added explanations on the static calculation in section 3.3, line 257-262, and provided an example of the static calculation for the two-turbine array in line 263-270. Also, we have added the sentence in section 3.5 for mentioning these explanations, see line 358.

For mooring lines with a proportion of the length laid on the seabed, mooring tensions can be calculated from the catenary equation (see Equation 4), given knowledge of the positions of fairleads and anchors, as well as line length, mass density and elasticity of mooring lines. Similar calculations of the catenary mooring lines can be found in the study (ref6). In static calculations, a constant aerodynamic force is applied to the floater and drives the floater to move in the horizontal plane. The surge and sway offsets change fairlead positions and thus changes mooring fairlead tensions. The resulting mooring forces at fairleads are balanced with this thrust to reach the force equilibrium for each floater.

Consider a two-turbine array as an example. The unknown variables are surge (x) and sway (y) offsets for the two turbines. Here $x_1$ and $y_1$ are for turbine 1, and $x_2$ and $y_2$ for turbine 2. The fairlead position ($x_f$ and $y_f$) for each line is the sum of the offset and the initial position. Based on $x_f$ and $y_f$ the resulting mooring forces $F_x$ and $F_y$ are derived from the catenary equation. For the shared line, the mooring force is a function of the four variables ($x_1, y_1, x_2, y_2$). To keep force equilibrium, the sum of tension components in X-axis ($\sum F_x$ ) are balanced with the thrust force and tension components in Y-axis must balance each other. Consequently, we obtain 4 equations for the force components in X- and Y- axis for the two-turbine arrays to solve those 4 unknown offset variables.

4.  Why is static turbine spacing selected as an independent design driver/objective? Please explain.

Replay: thank you for this point. We consider the FOWT distance in the design process, because this distance significantly impacts the turbine performance, particularly due to the wake effect generated by the upstream turbine on the downstream one. The wake deficit reduces the wind speed on the downstream turbine and thus affects the power production of this turbine. The added turbulence intensity also causes higher vibrations in the structure, such as blades, and results in more structural fatigue damage. Furthermore, this FOWT distance determines the shared mooring line length, which directly influences the material cost of the mooring system.

This distance also reflects the wind-farm layout, where a smaller distance theoretically corresponds to a smaller environmental footprint. This paper utilizes the layout information rather than designing the layout. So, a broad range from 6 D to 12 D is used to encompass the FOWT distance that found in the current floating wind farm layouts. The design target of the shortest distance aims to develop a solution that minimizes environmental impact. We have added this target in section 3.4 line 408-409.

5.  How does the mooring design approach proposed in this study align with the current rules and regulations for mooring design of floating offshore wind turbines? Please comment.

Replay: thank you for this point. To address the limitations of existing linearized design method, this paper proposes a comprehensive design methodology for shared mooring line configurations. In developing this method, the current rules and regulations for mooring designs have been considered to ensure this design methodology consistent with these standards. For instance, the chain elasticity is determined by the chain diameter and grade-dependent factor according to the rule (ref7), rather than by the linear scaling based on the line weight as used in the current linearized design method (ref2).

The analysis method includes both quasi-static calculations and dynamic simulations. This approach aligns with the philosophy for analysis model that the choice of method should depend on the sophistication and analysis objectives (ref8). The quasi-static calculation is efficient for filtering the designs that satisfied the targets of lowest cost and shortest distance. Meanwhile, the dynamic simulation can consider the wake effect, the wave force, the seabed friction, and the line dynamics to provide a more accurate prediction.

We have also ensured compliance with these regulations for mooring designs by implementing design variables and constraints related to facility safety and mooring integrity. Since the drag embedded anchor can only withstand horizontal forces, the design input variable 'length ratio' and the design constraint 'lay-down length redundancy' ensure that a portion of the mooring line must be laid on the seabed to prevent the anchor from being lifted. The design constraint 'offset limit' filters the mooring configurations to ensure the safety of dynamic cables, as large offset may cause damage to the cables. In addition, the peak thrust force is applied to the floater to calculate the maximum offset and the line strength is controlled by the design constraints 'pretension ratio' and 'lay-down length redundancy', to maintain that the peak tension within the minimum breaking load. This also aligns with the rule (ref8) that requires tension to be calculated at the maximum offset condition when a quasi-static analysis model is employed.

6.    This study focuses on a pilot-scale floating wind farm with 2-3 floating wind turbines. How to generalize the findings and conclusions in this study to commercial-size floating wind farms, i.e., with increased farm size?

Replay: thanks for this point. This paper demonstrates the quasi-static design workflow for shared mooring line configurations, and selects two design objectives for dynamic simulations to investigate the influence of shared mooring lines. While the primary aim of the paper is to propose a comprehensive design method for shared mooring designs, the findings concerning the cost savings in the material cost, as well as the performance influence in terms of the power, the floater motion and the mooring tension fatigue can be scaled to the larger wind farms, but with certain limitations.

1.    The proposed design method incorporates realistic design variables and multiple design constraints to improve the resilience of shared mooring designs, compared to the existing linearization method. Therefore, the quasi-static design workflow can serve as an efficient tool for shared mooring designs of the commercial-scale floating wind farms in the early design phase.
2.    Cost savings in mooring material for shared mooring configurations can be scaled up with the increased size of wind farms, as the mooring material cost is determined by the number of lines and anchors used. Shared mooring configuration S3 for the three- turbine array is more effective in reducing the material cost compared to S2 for the two-turbine array. However, the quantified cost savings are based on an empirical model specific to mooring chains and drag embedment anchors.
3.    Two design goals, lowest cost and shortest distance, are proposed in this paper. Two designs can slightly increase the total power the floating array and the distance-driven design is more effective in power enhancement than the cost-driven one. This motivates the optimization design to maximize the power enhancement for the commercial-scale floating wind farms. Since this paper considers two most critical wind speeds and neglects the wind-wave misalignment, it is suggested to run simulation under diverse environmental conditions for the commercial-scale floating wind farms.
4.    Two designs can significantly increase floater motions and result in higher mooring tension damage. For commercial-scale floating wind farms, it is important to consider the trade-off between power enhancement and fatigue damage in shared mooring designs. Future studies are recommended to optimize the shared mooring designs in order to mitigate the mooring fatigue damage while maintaining a stable power improvement.

**Reference:**

[1] Connolly, Patrick, and Matthew Hall. "Comparison of pilot-scale floating offshore wind farms with shared moorings." Ocean Engineering 171 (2019): 172-180.
[2] Wilson, Samuel, et al. "Linearized modeling and optimization of shared mooring systems." Ocean Engineering 241 (2021): 110009.
[3] Mahfou, Mohammad Youssef, et al. "D1. 3. Public design and FAST models of the two 15MW floater-turbine concepts." (2020).
[4] Vigara, Fernando, et al. "D1. 2. Design Basis for COREWIND project." (2019).

[5] Faltinsen, Odd. Sea loads on ships and offshore structures. Vol. 1. Cambridge university press, 1993.
[6] Jonkman, Jason Mark. Dynamics modeling and loads analysis of an offshore floating wind turbine. University of Colorado at Boulder, 2007.
[7] DNVGL, "Offshore standard DNVGL-OS-E301 Offshore mooring chain." (2018).
[8] ABS, "Position mooring system."(2020).